

# Observations of stratospheric gravity waves in the tropics: can GNSS-RO extend the SABER climatological record?

Marwa Almowafy[1,2], Corwin Wright[1], and Neil Hindley[1]

[1]Centre for Climate Adaptation and Environment Research, University of Bath, Bath, UK
[2]Astronomy and Meteorology Department, Cairo University, Giza, Egypt

**Correspondence:** Marwa Almowafy (ma3091@bath.ac.uk)

**Abstract.**

The quasi-biennial oscillation (QBO) is the most important phenomenon in the tropical stratosphere. It is mainly driven by small-scale gravity waves. Still, the representation of QBO in models is challenging because small-scale gravity waves are not well resolved in the models and the majority of the parametrization schemes are limited to vertical propagation only of

gravity waves. One solution to this is to use high-resolution satellite observations to understand the gravity wave (GW) forcing on the QBO. However, the results can vary from one observation to another due to the unique observational filter of each instrument. Here we investigate how these differences in the observational filters between SABER and GNSS-RO satellite measurements affect our ability to capture the interactions between GWs and the QBO. To test this, we sample temperatures from the high-resolution GEOS model as if they were observed by SABER and GNSS-RO and estimate synthetic GW potential

energy ($E_p$) observations. We then systematically vary the viewing angle and the vertical and horizontal resolutions of the instruments to determine which aspects have the most significant effect on the observed GW $E_p$. This allows us to understand how the observational filter of each instrument influences the observation of GW-QBO interaction and if we can bring the two observations close enough to get nearly the same results. Our results reveal that vertical resolution is the most significant factor deriving the differences between the results of both instruments. By adjusting the vertical resolution of GNSS-RO temperatures

to match that of SABER, we found that the GW $E_p$ and vertical wavelength measurements from both instruments could be brought into very close agreement.

This study not only focuses on the importance of selecting appropriate observational methods for gravity wave research but also highlights the potential of GNSS-RO to extend the long-term studies of GW interaction with the QBO that has been carried out by SABER for more than 23 years, especially as SABER approaches the end of its operational lifespan. Our

findings contribute to a more comprehensive understanding of GW observations in the tropics and provide a foundation for future applications using merged GNSS-RO observations.

## 1   Introduction

The Quasi-Biennial Oscillation (QBO) is the dominant mode of variability in the tropical stratosphere, characterized by alternating and descending eastward and westward winds over a cycle of approximately 28 months (Baldwin et al., 2001). This



oscillation is mainly driven by atmospheric waves, with gravity waves (GWs) playing a significant role in westward forcing, particularly those with short vertical wavelengths (e.g. Anstey et al., 2022; Kim et al., 2024). Numerous observational studies have demonstrated that GWs with vertical wavelengths of less than 10 km are essential for driving the QBO (e.g. Ern et al., 2014; Vincent and Alexander, 2020). These waves deposit momentum into the stratosphere through dissipation, which reduces the wave momentum flux and causes the zonal wind to shift towards the wave phase speed, leading to the descending shear

zones of the easterlies and westerlies (Dunkerton, 1997).

    Accurate measurement of these gravity waves is crucial for understanding the dynamics and variability of the QBO. Two common remote sensing observations used for such studies are SABER and GNSS-RO, both of which provide long-term global coverage of the atmosphere. SABER has been operational since 2002 (e.g. Remsberg, 2008), while the earliest GNSS-RO mission used for studying gravity waves was using Challenging Minisatellite Payload (CHAMP), which was launched in

2006 (e.g. Poli and Joiner, 2003; Wickert et al., 2002).

    Although both instruments are capable of measuring small-scale gravity waves and their interaction with the QBO, in practice they capture different parts of the gravity wave spectrum due to their varying sensitivities, observational geometries, and data retrieval processes. These differences are known as the observational filters of the two instruments, a concept introduced by Alexander (1998). The effect of the observational filter is a critical factor influencing the estimation of gravity wave properties,

as has since been highlighted in previous studies (e.g. Trinh et al., 2015; Wright et al., 2016).

    No existing observational technique to date can provide both the global coverage and the spectral and temporal resolution necessary to capture the full spectrum and geographic distribution of gravity waves (Alexander, 1998; Preusse et al., 2009; Alexander et al., 2010). This limitation is important because gravity wave properties can vary depending on their spectral characteristics (Wright et al., 2016). Therefore, understanding how the observational filters of different instruments influence

gravity wave measurements is essential for integrating observations from various sources and gaining a more comprehensive view of the gravity wave spectrum (Preusse et al., 2000; Wu et al., 2006).

    The timeliness and importance of this study are underscored by the fact that SABER instrument, originally designed for a two-year mission, has been operational since January 2002 (Brown et al., 2006) and is now in its 23rd year of operations. As SABER ages and eventually becomes non-operational, GNSS-RO is expected to continue to provide long-term observations.

Accordingly, in this study we compare the results from both instruments, allowing us to explore the potential for extending the work carried out in previous SABER-driven studies of GW driving of the QBO studies using GNSS-RO data after SABER's operational life ends. Our primary goal is thus to examine the differences and similarities between GNSS-RO and SABER in capturing gravity waves and their interactions with the QBO, and to quantify the differences between the instruments caused by the observational filter differences.

To do so, we conduct three types of analyses on the data: (1) we apply the same analysis method to datasets from both GNSS-RO and SABER, allowing us to investigate whether the observed differences in gravity wave potential energy measurements are due to observational limitations or analytical choices, (2) we perform independent analyses on synthetic data designed to approximate measurements made by both GNSS-RO and SABER to pinpoint the source of observational discrepancies, and (3) we then apply the insights from the synthetic data analysis to the observational data to assess how well the synthetic results





align with real-world observations and whether they explain the observational filter differences between the two instruments. While previous studies have compared GW $E_p$ from SABER and COSMIC (e.g., Wright et al., 2011, 2016), this is the first study to include several merged GNSS-RO missions and compare them with SABER.

Our study focuses on the QBO region (defined as $10°$S to $10°$N), where gravity wave interactions with the QBO have been analyzed individually using GNSS-RO and SABER, but not directly compared between the two instruments. This comparison

is critical, as it allows us to examine the impact of each instrument's observational filter on capturing GW-QBO interactions and explore the potential for GNSS-RO to extend SABER's long-term observations, contributing to future research on GW-QBO dynamics.

The paper is structured as follows: Section 2 describes the two observational instruments, SABER and GNSS-RO, along with the characteristics of the GEOS model dataset, which we use to simulate temperatures as retrieved by both instruments.

Section 3 outlines the methods used to extract gravity wave signatures and their wave parameters. In Section 4, we present the findings from the direct comparison of observations, followed by quantification of key observational filter parameters in Section 5, where we identify which factors most significantly impact GW $E_p$ and vertical wavelength estimates, then the application of the model results on observations in Section 6. Finally, Section 7 provides a summary and discussion of the findings, and Section 8 concludes the main points of the study.

## 2  Data

### 2.1  ERA5

The ERA5 dataset is the fifth generation of reanalysis products from the European Centre for Medium-Range Weather Forecasts (ECMWF), generated using version 41r2 of their Integrated Forecasting System (IFS) with Four-Dimensional Variational (4DVar) data assimilation to combine model forecasts with observational data. The dataset includes hourly estimates of atmo-

spheric variables such as temperature, humidity, wind, and pressure, as well as derived quantities like precipitation and cloud cover, at a horizontal resolution of approximately 31 km. To prevent wave reflections at the model top, two artificial sponge layers are employed in the IFS, with a weak sponge starting at 10 hPa and a stronger one at 1 hPa (Hoffmann et al., 2019; Hersbach et al., 2020).

In this study, we use zonal wind data averaged at the tropics defined here as: $10°$S $– 10°$N. We analyze ERA5 3-hour data

at $1.5°$ horizontal sampling on standard pressure levels, focusing on 16 years (2007-2022). The data are daily averaged. The selection of the spatial and temporal coverage is to ensure consistency with other datasets.

### 2.2  GNSS-RO

Global Navigation Satellite System Radio Occultation (GNSS-RO) is a remote sensing technique that utilizes signals originally from GPS and currently from GNSS satellites including GLONASS, GALILEO, and BeiDou to study the Earth's atmosphere

(Leroy et al., 2023). The effectiveness of GNSS-RO was first demonstrated by the Global Positioning System Meteorology





(GPS/MET) experiment between 1995 and 1997 (Kursinski et al., 1997; Rocken et al., 1997), and continuous RO observations began with the CHAMP satellite in 2001 (Poli and Joiner, 2003; Wickert et al., 2002).

For this study, GNSS-RO data were obtained from three major global retrieval centers: the COSMIC Data Analysis and Archive Center (CDAAC) at the University Corporation for Atmospheric Research (UCAR), the NASA Jet Propulsion Lab-
oratory (JPL), and the Radio Occultation Meteorology Satellite Application Facility (ROM SAF). The data, which have been made publicly available through the AWS Open Data Registry recently, provide long-term, high-quality, and standardized GNSS-RO profiles, supporting consistent atmospheric monitoring. Detailed information about the data, including a description and details of accessibility, is provided by Leroy (2022).

The merged dataset includes vertical profiles of temperature, pressure, and humidity, along with derived parameters such as
geopotential height and refractivity. It offers global coverage from the surface to the upper atmosphere (0 – 50 km). The nearly dry stratosphere allows for the direct use of dry temperature profiles from satellite measurements.

GNSS-RO data provide vertical resolution superior to that of many other satellite-based remote sensing techniques. This high resolution greatly enhances the study of atmospheric gravity waves (Wright et al., 2011; de la Torre and Alexander, 2005; Schmidt et al., 2016; Tsuda et al., 2000; Wang and Alexander, 2010). Temperature profiles have a vertical resolution ranging
from 0.5 km in the lower troposphere to 1.4 km in the stratosphere (Kursinski et al., 1997; Wright et al., 2011). Dry temperature data from GNSS-RO measurements have a precision of approximately 0.5 K in the lower stratosphere, though errors increase at altitudes above 35 km due to lower air density (Wang and Alexander, 2010; Tsuda et al., 2011).

In this study, we use dry temperature profiles from GNSS-RO measurements collected during 13 satellite missions (GP-S/MET, GRACE, SACC, CHAMP, COSMIC1, TSX, TDX, C/NOFS, MetOp, KOMPSAT-5, PAZ, Geoopt, and Spire) over the
tropics ($10°$S – $10°$N) for the period from 2007 to 2022 (Leroy et al., 2023). This allows for merging a large number of profiles per day, reaching up to 10000 profiles daily.

## 2.3   SABER

Sounding of the Atmosphere using Broadband Emission Radiometry (SABER) is one of four instruments on NASA's Thermosphere, Ionosphere, Mesosphere Energetics and Dynamics (TIMED) satellite, and was designed to measure infrared emissions
in the Earth's atmosphere. Launched in 2001, SABER has been continuously providing data since January 2002 and measurements are still ongoing at the time of writing (e.g., Mlynczak, 1997; Russell et al., 1999; Yee et al., 2003). The dataset is publicly accessible through NASA's data archives.

The instrument operates in a limb-viewing geometry. Kinetic temperature profiles span altitudes from 15 to 120 km, with a precision of around 0.8 K in the stratosphere (Remsberg, 2008). SABER provides continuous near-global coverage with
constant measurements between $50°$S and $50°$N year-round, and extending to either $80°$S or $80°$N during an alternating 60-day yaw cycle.

SABER provides approximately 2,200 atmospheric profiles globally per day, with a vertical resolution of about 2 km, along-track profile spacing between 200 and 550 km, and 50 km across the line of sight (LOS). The line of sight (LOS) is $90°$ off track (Mlynczak, 1997). SABER has been widely used in atmospheric studies, particularly in research on gravity waves, atmospheric



tides, and thermospheric dynamics (e.g., Krebsbach and Preusse, 2007; Preusse et al., 2009; Ern et al., 2011; Wright et al., 2016; Liu et al., 2017). We used daily temperature data over the tropics from 2007 to 2022.

## 2.4    GEOS

The Goddard Earth Observing System (GEOS) is a high-resolution global non-hydrostatic atmospheric general circulation model (Putman et al. (2014); Norris et al. (2015)). It uses a cubed-sphere horizontal grid with 2880 cells per edge, resulting
in a total of 17,280 horizontal grid cells. This configuration provides a global grid resolution of approximately 3.125 km. The vertical grid of the model consists of 181 hybrid sigma-pressure levels, spanning from the surface to an altitude of about 0.01 hPa (approx. 85 km). The lowest level is situated 18 m above the surface, and a sponge layer is applied to the top 18 levels (i.e. from 0.3 to 0.01 hPa) to prevent artificial reflections (Stephan et al., 2022).

     The vertical resolution is approximately 200 m or less below 800 hPa ($\sim 2$ km), around 500 m near 600 hPa ($\sim 4$ km), 1 km
near the tropopause (between 8 km at the poles and 18 km at the tropics), and about 2 km near the stratopause ($\sim 50$ km). This level of vertical resolution and altitude range enables the resampling of GEOS data to closely match the retrieved temperatures from GNSS-RO and SABER, as the required vertical resolution of GNSS-RO is 1.2 km and for SABER is 2 km within the lower stratosphere.

     We use model data from the DYAMOND-WINTER run, sampled to match the retrieved temperature data from SABER and
GNSS-RO as described in Section 5. The simulation was initialized with a common atmospheric analysis from the European Centre for Medium-Range Weather Forecasts (ECMWF) and ran for 40 days, with specified sea-surface temperatures updated every 7 days, as part of the atmosphere-only experiments (Holt et al., 2017; Stephan et al., 2022).

     Our study concentrates on the first week of the simulation. This period is chosen as since the model is free-running, we expect significant divergence from the true state at run lengths beyond this (see (e.g., Lear et al., 2024)).

## 145    3    Methods

### 3.1    Background Removal and Spectral Analysis

We first analyze the time series of gravity wave potential energy (GW $E_p$) derived from GNSS-RO and SABER temperature observations over the tropics ($10°$S - $10°$N) in the lower stratosphere between 20 and 40 km altitude.

     The GNSS-RO raw data undergoes several preprocessing steps to prepare it for gravity wave analysis. First, we extract the
required variables from the data files and restructure them into a unified daily data format of profiles per day by altitude. We then select altitudes between 0 and 40 km, to ensure the accuracy of the GNSS-RO data, interpolating the data to a vertical resolution of 0.1 km across all variables including temperature, latitude, longitude, and time. A data cleaning process is applied, filtering out temperatures below 100 K or above 400 K, data beyond the latitude range of $-90°$ to $90°$ and longitude range of $-180°$ to $180°$, and any data above 40 km.





SABER raw data requires a slightly different approach. We first apply the same unified structure used for GNSS-RO, orga-
nizing it as daily Profiles × Altitude for the study period (2007-2022). The altitude range for SABER data are set between 0
and 120 km, and the data are interpolated to a 0.5 km vertical spacing, to avoid oversampling the original data. Post-quality
control for temperatures, latitude, and longitude is applied similarly to GNSS-RO.

After the quality control check, both datasets are trimmed to the study region (20 – 40 km, ±10° latitude). To ensure
consistency in the comparison between GNSS-RO and SABER, the GNSS-RO data are at this stage downsampled to a 0.5
km vertical spacing to match SABER; note that this still represents an oversampling relative to the expected resolution of the
observations.

To identify gravity wave signatures from the temperatures we then apply a new method to the temperatures to remove large-
scale variations including Kelvin waves, tides, or planetary waves, which we collectively treat as a large-scale background.
This method was adapted from our previous work using meteor radar data to estimate bulk winds (Hindley et al., 2022). We
assume that perturbations from this background represent gravity wave signatures.

Previous work has employed a variety of analysis methods to extract gravity wave signatures from temperature observations.
One common approach involves applying a vertical filter with a chosen cutoff vertical wave number or wavelength, (e.g., de la
Torre and Alexander, 2005; Tsuda et al., 2000; Hindley et al., 2015). Such a method does not effectively remove Kelvin waves
signals from the temperature fluctuations, since the vertical wavelength of gravity and Kelvin waves could be similar (e.g.,
Holton et al., 2001); since these are common and strong in magnitude in the tropics, such an approach therefore cannot be
used here. Schmidt et al. (2008) used a modified Gaussian window to separately filter tropospheric and lower stratospheric
profiles, however, this method shows some discontinuities at the tropopause which might introduce artificially enhanced GW
activity. Finally, Wang and Alexander (2010) binned temperature profiles within each day to a 15° × 10° longitude and latitude
resolution, then applied a Stockwell transform (S-Transform) (Stockwell et al., 1996) to remove zonal wavenumbers 0–6.
This method is dependent on data coverage and the number of available measurement points per bin to accurately represent
the background. It can introduce some artifacts in the perturbations particularly near the tropopause. GNSS-RO dataset is
heterogeneous in coverage and especially sparse around the equator for the period before 2019.

In this study, we instead use a weighted sine fitting approach to identify and remove the large-scale background from
our observations. This weighting approach has significant advantages in terms of more robust derived values over a binning
approach Hindley et al. (2022).

This approach works by fitting sinusoids for zonal wave numbers 0-9 to separate PWs and large-scale background conditions
from our data. To do this, we use a daily weighted sine fitting routine Hindley et al. (2022) where weightings in latitude,
altitude, and time are derived from Gaussian functions with Full Width Half Maxima (FWHMs) set to 4 degrees, 2 km, and
1 day, respectively, and stepped in increments of 2 degrees (in latitude), 1 km (in altitude), and 1 day (in time). Using this
approach, we then assume that waves with zonal wavenumbers from 0-9 represent the background and are subtracted from the
total temperature to obtain the residual temperatures. Figure 1 illustrates the steps of how the method works.

We then apply the S-Transform to the individual temperature residual profiles to derive the amplitudes and vertical wave-
lengths of GWs (e.g., Alexander et al., 2008a; Wright et al., 2011; Wright and Gille, 2013; Wright and Hindley, 2018). The




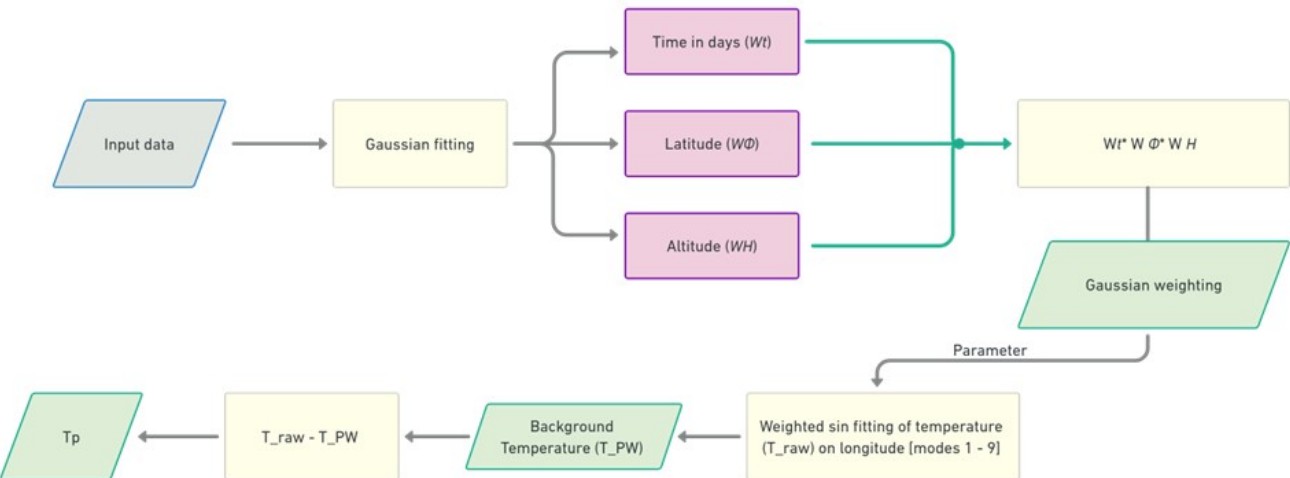

**Figure 1.** A flowchart demonstrating the steps of the background removal method from the retrieved temperature data to the temperature residuals where the S-Transform is then applied to.

S-Transform provides time-localized frequency information. Since the S-Transform is based on a Fast Fourier Transform (FFT), to avoid any wraparound at the vertical ends of the data we apply a zero-padding of 20 km at the top and bottom of the profile before applying the S-Transform (Wright et al., 2010). Due to the short vertical range of the GNSS-RO data, we also impose a vertical wavelength detection limit of 20 km. As part of the post-processing, we focus particularly on small-scale gravity waves, excluding any with vertical wavelengths greater than 12 km.

The derived gravity wave temperature amplitudes are used to calculate the GW Potential Energy ($E_p$) as:

$$E_p = \frac{1}{2}\frac{g^2}{N^2}\left(\frac{\overline{T'}}{T_0}\right)^2 \tag{1}$$

where, $g$ represents the acceleration due to gravity, $N$ the buoyancy frequency derived from measured background temperatures, and $T'$ and $T_0$ represent temperature amplitude and background, respectively.

For our subsequent analyses, we use daily averages of all the profiles measured within each day. The same analysis has been
performed on both GNSS-RO and SABER datasets.

## 3.2 Definitions

Before we describe our main results, it is important to define some key terms that will be used extensively throughout the study.

Various studies on the QBO have chosen specific altitudes for their analysis, such as 30 km by (Nath et al., 2014), $\sim$ 24 km (30 hPa) by (Osprey et al., 2016), and 28 km by (Ern et al., 2014). Given that the core of QBO variability occurs between
20 and 30 km, here we use the zonal wind from ERA5 data at 25 km over the latitudinal band of $\pm10°$ around the equator to define the QBO phases and cycles. A QBO cycle is defined in this study as the period corresponding to the distance between 2



successive eastward peaks of the zonal wind. The eastward (positive) and westward (negative) wind phases are referred to as the EQBO and WQBO phases, respectively, as shown in Figure 2(b).

The colour plot shown in Fig.2(c) shows a time-height cross-section of an arbitrary QBO cycle with a period of 28 months. The time series plot shows the zonal wind at 25 km, with the positive part corresponding to the eastward phase of the QBO (EQBO) and the negative part corresponding to the westward phase (WQBO). The profile shown in Figure 2(a) illustrates how we define the eastward and westward wind shears. The eastward wind shear is defined as the change of the wind with altitude from westward to eastward, and vice versa for the westward wind shear.

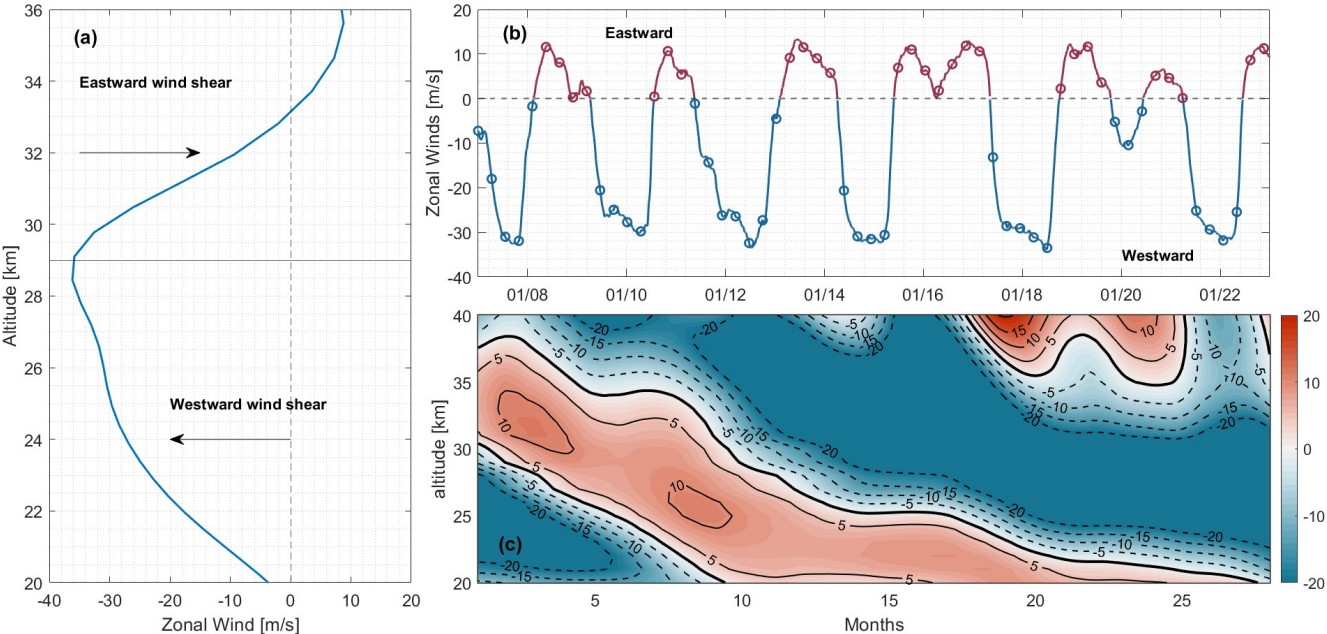

**Figure 2.** Schematic representation of the QBO terminology through the study. Panel (a) shows the Eastward and Westward wind shear. Panel (b) shows the EQBO and WQBO, and Panel (c) is one arbitrary regular QBO cycle from the time series that shows the westward phase (blue) with the dashed contour lines and the eastward phase(red) with the solid contour lines. the thick black contour lines show the zero-wind line.

## 3.3 QBO cycles

The chosen period (2007 - 2022) encompasses five full QBO cycles. This period include the only two disruptions recorded in over 60 years of QBO data, specifically the major 2016 (e.g., Osprey et al., 2016) and 2019 (e.g., Anstey et al., 2022) disruptions. Since these two QBO disruptions are quite different in character, we consider each individually in our subsequent analyses.

Figure 3 shows the ERA5 zonal mean zonal winds in the tropics within the study period. We first divide the five cycles into 220 two categories, specifically 'regular' and 'disrupted'. A 'regular' cycle is a cycle of the period length of in average 28 months,




while the disruptions are defined according to both literature as well as the cycles where a weakening of the eastward winds in the lower stratosphere occur interrupted by the westward wind (Osprey et al., 2016). Note that the regular QBO cycles vary in length and sometimes in structure. For example, the eastward wind shear is stronger at the end of 2009 indicated by the steep slope of zero-line wind, and the eastward wind is stalled at the beginning of 2009.

We compared ERA5 zonal winds to Singapore sonde winds and no significant differences are found between both datasets in their regions of vertical overlap. Based on this, we choose to use ERA5 for better coverage over the altitude ranges observed by GNSS-RO and SABER.

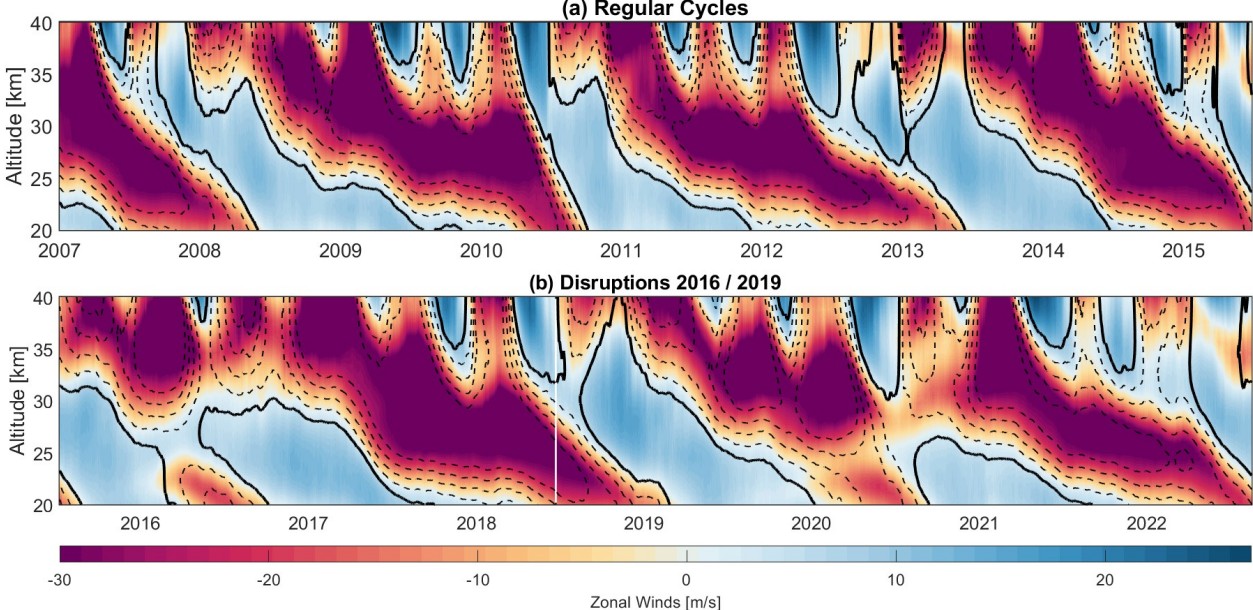

**Figure 3.** Time-Height cross-section of the ERA5 zonal winds at the tropics for the period 2007 - 2022. Regular cycles (a) and the 2016 and 2019 QBO disruptions (b). The dashed contour lines and the purple colour represent the westward (negative) winds, the blue colour represents the eastward (positive) winds, and the solid lines indicate the zero-wind lines.

## 4    SABER and GNSS-RO Observation Results

In this section, we compare both quantitatively and qualitatively the features of the GW signature derived from SABER and
GNSS-RO observations. We carry out this comparison in terms of GW $E_p$ and vertical wavelength $\lambda_z$; these two metrics are chosen as unlike more heavily-derived parameters such as momentum flux, they can be measured easily using both instruments.

     Before proceeding with analysis and comparison of both datasets, we first highlight the differences in observational filters between both instruments and how these differences may introduce discrepancies and biases in the results, potentially leading





to the two instruments preferentially measuring different parts of the true geophysical GW spectrum. These main differences
are:

- SABER measures the atmospheric state along a regularly-spaced track, with a horizontal sampling spacing alternating between approximately 300–500 km between adjacent profiles at the height 30 km, arising as a result of the instrument's vertical scanning pattern (see e.g. Remsberg (2008)). In contrast, GNSS-RO provides profiles pseudorandomly located in both time and space, resulting in non-uniform coverage across different regions of the globe (Anthes et al., 2008).
This difference in coverage leads to differences in the observed waves. For example, with SABER, we can estimate the horizontal wavelength ($\lambda_h$) of the wave given the distance between the adjacent profiles in time; however, accurate measurements of the $\lambda_h$ are difficult or impossible for GNSS-RO because the profiles are not regularly spaced. It was possible to estimate the GW momentum flux (MF) and $\lambda_h$ only at the early missions of GNSS-RO where the satellites were closely spaced, and this has been done in (e.g., Wang and Alexander, 2010; Faber et al., 2013), but this is not an
option in the more general case we consider here.

- Due to the nature of the instrument's temperature retrieval, the GNSS-RO measurements become increasingly dominated by *a priori* estimates of the atmospheric state with increasing height, making the data unreliable above about 40 km. However, below 40 km, GNSS-RO data maintains a high level of quality. In contrast, SABER data performs well in the 20–90 km altitude range but is less reliable at lower altitudes. Therefore, the most reliable range for comparison between
these two datasets is from 20 to 40 km (Fan et al., 2015). This sensitivity to specific altitude ranges affects the estimation of the GW $E_p$ from both instruments, with more reliability of GNSS-RO GW $E_p$ results at lower altitudes than SABER and vice versa for higher altitudes ($> 40$ km) (e.g. Wright et al., 2011, 2016).

- SABER and GNSS-RO detect gravity waves differently due to their distinct geometries and line-of-sight (LOS) orientations relative to the wavefront. For GNSS-RO, any given wave is observed from a random direction, which can lead to
amplitude degradation and phase shifts across individual profiles (Alexander et al., 2008b). Since the LOS can intersect the wavefront at different angles, this results in apparent horizontal and vertical wavelengths that can differ from the true state, and wavelengths that can be distorted based on the relative orientation of the observation to the wave (Alexander et al., 2008b). For SABER meanwhile, the latitude position of the measurement and orbital node from which the measurement is taken determines the angle from which each wave is observed, leading to a more consistent product with
quite different direction biases to GNSS-RO, which can bias vertical wavelength and amplitude estimation (Trinh et al., 2015; Schmidt et al., 2016; Wright and Hindley, 2018).

- GNSS-RO provides high vertical resolution, due to the precise bending angle measurement, allowing for measuring waves within the range $2\text{km} \leq \lambda_z \leq 25\text{km}$ (Wright et al., 2016). The horizontal resolution is also high, typically on the order of tens of kilometers, due to the large number of GPS satellites and the wide distribution of receiving stations (
$\sim 270\text{km} \leq \lambda_h \leq 2000\text{km}$) (Wright et al., 2011, 2016; Wright and Hindley, 2018). SABER on the other hand provides longer wavelengths $4\text{km} \leq \lambda_z \leq 25\text{km}$ and $400\text{km} \leq \lambda_h \leq 2000\text{km}$.




Given such differences, we expect GW properties measured by these two instruments to deviate from one another. To compare them as fairly as possible, in this study we use the same analysis method on both datasets and in doing so limit the analysis to GWs with vertical wavelengths ≤ 12 km. In subsequent Sections, we will compare SABER and GNSS-RO measured GW parameters first using a simple periodogram analysis, after which we will compare GW $E_p$ and $\lambda_z$ estimated from both instruments during different QBO cycles.

### 4.1 Periodogram

We first present results from a periodogram analysis of GW $E_p$ derived from GNSS-RO and SABER data and ERA5 zonal winds, focusing on the tropics at an altitude of 25 km.

To permit a simple visual comparison, in our analysis we scale the power spectra resulting from this analysis to twice the total variance of each dataset for comparability. Figure 4 shows the resulting scaled periodogram, with the ordinate representing the scaled power spectra and the abscissa the periods of the inferred signals. Two scenarios were analyzed: the first covers the period without disruptions, from 2007 to 2015 (panel (a)), and the second includes the disruption period, spanning from 2007 to 2022 (panel (b)).

Figure 4(a) shows in black the periodogram measured for ERA5 zonal winds in the QBO region during the time where only regular cycles appear (corresponding to the periods shown in Fig. 3(a)). Periodograms derived from GW $E_p$ as observed by GNSS-RO and SABER are overlaid in red and blue respectively. The dominant peak in the zonal wind data occurs at 2.44 years (approximately 29.35 months), which is slightly longer than the typical QBO cycle period of about 28 months. This peak is 90% larger than the second-highest peak, indicating that the primary temporal characteristic of the zonal wind dataset is an oscillation with an average period of roughly 29 months. The peak does not align exactly with 28 months due to the irregularity of QBO cycles, which can extend beyond 28 months, and due to the short length of our data record relative to the long-term variability of the QBO.

The periodogram of GW $E_p$ from SABER (blue line) exhibits two prominent peaks: a primary peak at a period of 1 year and a secondary peak corresponding to the typical QBO period. The primary peak indicates an annual variability in SABER GW $E_p$, consistent with earlier findings where gravity waves exhibited strong annual periodicity (Preusse et al., 2009; Zhang et al., 2012; Shuai et al., 2014). The secondary peak at 2.44 years confirms that SABER GW $E_p$ is modulated by the QBO, aligning with previous studies such as Ern et al. (2014), which demonstrated that GW momentum flux with short vertical wavelengths ($\lambda_z \leq 12$ km) interacts with the QBO. The annual peak is more dominant than the QBO peak, even when focusing on short vertical wavelengths. This indicates that the GW spectra captured by SABER is dominated by a strong annual cycle rather than by interactions with the QBO.

The periodogram of GW $E_p$ derived from GNSS-RO data presents a different pattern than SABER, featuring a broad primary peak at 2.72 years (around 32 months) and a smaller secondary peak at 1.25 years. The primary peak aligns closely with the zonal wind peak, while the annual peak reflects the seasonal variation of GW $E_p$. The 1.25-year peak remains unclear, though it may be related to shorter QBO cycles, which will be explored later.





The peaks and the shape of the periodogram changed when selecting longer period. This is true of all three datasets as shown in Figure 4(b). The reduced power of zonal winds is a mathematical consequence of the additional periodicity introduced by the disruptions, which occurs with periods over 28-29 months. While the dominant peak from the ERA5 zonal wind remains unchanged, a second peak of nearly equal power appears at approximately 2.92 years ($\simeq$ 34.6 months). Additionally, there is a third peak at 3.6 years ($\simeq$ 43.2 months) and two double peaks of equal power at 1.8 and 2 years. This variability can be partly

explained by changes in the apparent length of the QBO periods during these disruptions. This result indicates that the time series of the zonal wind exhibits a superposition of short periods of $1.8 - 2$ years and a longer period of approx. 2.3 years, as well as a longer period of the disruptions which is about 3.6 years.

Although the GW $E_p$ from SABER shows a similar overall structure to the ERA5 zonal wind periodogram, the dominant peak remains at 1 year, with stronger power spectra compared to Panel (a). The second most prominent peak aligns with

the regular cycle of the QBO. This confirms that the primary feature of the GW spectra observed by SABER is its annual variability, reflecting a pronounced seasonal cycle in the GW energy.

The GW $E_p$ from GNSS-RO shifts with the change in analysis period to include the disruptions, now showing a dominant peak at 3 years (36 months), which remains stronger than the annual peak. The increased power at around 3 years suggests a stronger interaction between GW spectra captured by GNSS-RO and the QBO..

A persistent peak at 1.25 years is evident in all three datasets, suggesting a strong interaction between gravity waves (GW) and the quasi-biennial oscillation (QBO) in both SABER and GNSS-RO data. In zonal wind, the peak at 1.25 is not as strong as the peaks at 1.8 and 2 years, however, it is visible in the three datasets; ERA5 zonal winds, GNSS-RO, and SABER GW $E_p$. This is possibly due to the fact that zonal wind during the disruption apparently exhibits connected short periods of the QBO with a length of approximately 1.25 years.

It is well known that small-scale gravity waves contribute to the disruption of the QBO (Li et al., 2023). This periodogram analysis suggests that GNSS-RO, with its higher sensitivity to shorter vertical wavelengths, finer vertical resolution, and broader altitude coverage, captures these small-scale waves and their effects more effectively. In contrast, SABER is more sensitive to longer wavelengths, perhaps leading to undersampling of the shorter scales. This difference may explain why the effect of the two disrupted QBOs is clearly visible in the periodogram of GW $E_p$ from GNSS-RO but not as distinctly in the SABER data.

The results further suggest that the ability of GNSS-RO to capture gravity wave spectrum changes linked to the QBO better than SABER may mainly be due to differences in their respective observational filter.

Although the periodogram method is highly sensitive to the length and variability of the input data, it remains a valid approach that confirms previous findings about small-scale GWs playing a major role in driving the QBO (e.g., Wang and Alexander, 2010; Anstey et al., 2022; Nath et al., 2014; Tsuda et al., 2011). Instruments with longer vertical wavelengths, such

as AIRS data, for example, do not show any QBO signal (Hindley et al., 2019). These results underscore the importance of comparisons between SABER and GNSS-RO as a means of highlighting the instrument's relative sensitivity to different scales of gravity waves and of identifying the influence of their observational filters in capturing GW-QBO interaction.





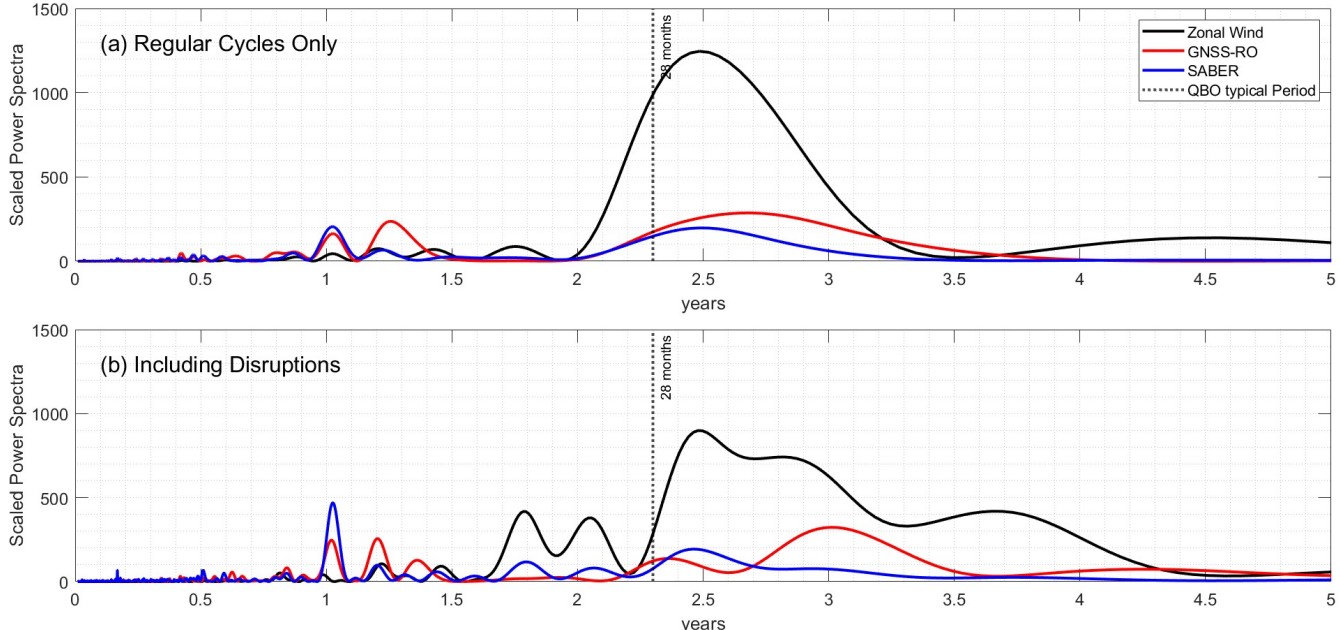

**Figure 4.** Periodogram of the ERA5 zonal mean zonal winds (black), GW $E_p$ from GNSS-RO (red), and SABER (blue). The theoretical period of the QBO is indicated by the dashed line. Panel (a) is the period from 2007 to 2015, where no disruption occurs, and panel (b) is for the whole period (2007 - 2022), including 2016 and 2019 disruptions. The x-axis is the time in years. the y-axis is the scaled power spectral density.

## 4.2 Potential Energy from GNSS-RO and SABER

We next proceed to examine the key differences between GNSS-RO and SABER in capturing the GWs-QBO interaction.
Specifically, in this section we examine the difference in GW $E_p$ measured by the two instruments, and how this is modulated by the QBO.

Figure 5 shows the GW $E_p$ difference between GNSS-RO and SABER averaged over the latitudes $\pm 10°$ during different QBO cycles. For simplicity, and since the GW $E_p$ from both instruments doesn't vary significantly from one regular QBO cycle to another, we chose to focus on the first regular QBO cycle within the study period as a representative example of the
three regular cycles. In contrast, the 2016 and 2019 disruptions are presented separately due to their distinct characteristics. Our results exhibit several key features:

1. GNSS-RO exhibits higher GW $E_p$ than SABER in all cycles, as shown by Figure 5(a). This difference is on average of order 2 $\mathrm{Jkg}^{-1}$ and reaches a maximum of 5 $\mathrm{Jkg}^{-1}$ at lower altitudes (20 – 25 km). The difference decreases with altitude until it almost vanishes at altitudes above 35 km.

2. The most evident dissimilarity in GW $E_p$ is found at lower altitudes and during the transitions between QBO phases, such as from EQBO to WQBO and vice versa. This is attributed to the varying sensitivity of each instrument to specific



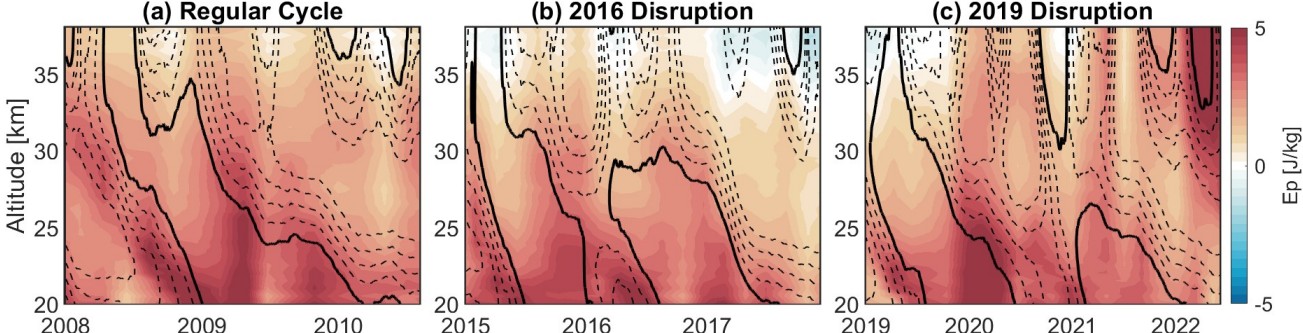

**Figure 5.** Time-Height cross-section of the difference in GW $E_p$ between GNSS-RO and SABER (color map) overlaid by the zonal winds at the QBO region shown in contour lines, with solid lines represent zero-wind line and dashed lines represent the westward winds with a spacing of 5 $ms^{-1}$ during one regular cycle (a) the 2016 disruption (b) and the 2019 disruption (c)

altitude ranges (Tsuda et al., 2009; Ern et al., 2018). The GNSS-RO exhibits higher $E_p$ at the phase transitions and during the eastward phase because of the variation of the GW convective sources around the equator at the northern and southern hemispheres. This is in agreement with previous studies showing that most of the GW activity is found at the transitions between QBO phases and due to the inter-hemispheric variability e.g., (De la Torre et al., 2006; Nath et al., 2014; Luo et al., 2021).

3. During the 2016 disruption, the GW $E_p$ difference between GNSS-RO and SABER is smaller above ∼30 km compared to the other two cycles and even reverses at an altitude of around 34 km, where GW $E_p$ from SABER slightly larger than that of GNSS-RO.

4. The difference is smaller during the WQBO phase. Notably, GW $E_p$ tends to diminish or be filtered out, particularly from SABER, above the zero-wind line. This is consistent with previous findings by Ern et al. (2014).

5. During the 2016 disruption, the difference is minimal above 30 km. This is possibly due to the the decreased GW activity during 2016 disruption (Luo et al., 2021; Li et al., 2023).

## 4.3 Vertical wavelength from GNSS-RO and SABER

Next, we investigate how the observational filters of each instrument affect the vertical wavelength of the observed gravity waves.

During 'typical' QBO cycles, both GNSS-RO and SABER display a consistent pattern of longer vertical wavelengths at higher altitudes and shorter ones at lower altitudes. This is consistent with our understanding of GW propagation, as shorter vertical wavelengths are typically filtered out by the wind at lower altitudes, while larger vertical wavelengths are less affected by dissipation and, therefore, tend to propagate higher (e.g., Yiğit and Medvedev, 2016; Heale et al., 2018). However, we find



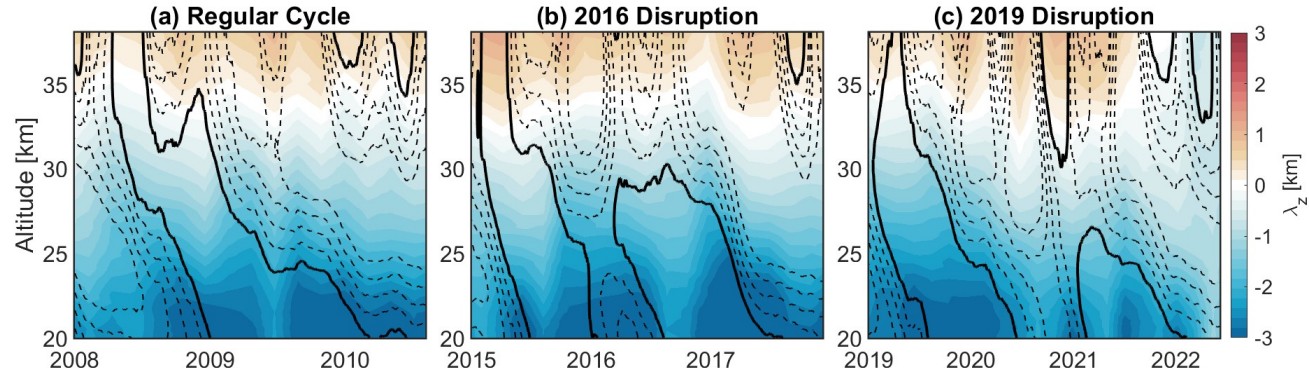

**Figure 6.** Same as Fig. 5 but for Vertical Wavelength ($\lambda_z$).

that the shorter wavelengths detected by GNSS-RO are associated with a larger range of $E_p$ compared to SABER. Notably, during the three QBO cases, vertical wavelength from GNSS-RO is minimum at the transitions from EQBO to WQBO and vice versa except at the end of the 2019 disruption. This trend is not observed in SABER.

## 5    Model Sampled Data

When we identified discrepancies between our results from SABER and GNSS-RO, we ascribed these differences to the limitations in each instrument's observational filter in Section 4. However, the exact causes behind these variations remain unclear - in particular, which specific elements of the observational filters contribute to these differences? In this section, we investigate the factors driving these discrepancies, to link them to measurable parameters. Additionally, we aim to gain a better understanding of how different the observational filters of these two instruments. Addressing these differences is crucial to

determining whether GNSS-RO data can be used to extend the study of GW-QBO interactions, building upon the extensive previous work using SABER.

To do this, we use one week of output (representing the period January 20–26, 2020) from a 40-day free-running DYAMOND-WINTER simulation carried out using the 3 km resolution GEOS model. These dates were selected as they are close to model initialisation on the $20^{th}$ of January, and thus represent a broadly realistic atmospheric state. Using this model output, we

produce synthetic estimates of temperatures in the model as they would be observed by SABER- and GNSS-RO-type instruments. The high resolution of the GEOS model allows it to capture a large fraction of small-scale gravity waves (Holt et al., 2016, 2017). The model is capable of reproducing a realistic QBO in (Holt et al., 2016), making it an appropriate choice for our study.

In Section 4, we discussed the observational filter differences between SABER and GNSS-RO. To identify the key factors

contributing to the discrepancies in GW $E_p$ results between the two instruments, we here sample GEOS temperatures to simulate the temperature retrievals of SABER and GNSS-RO, following the method from Wright and Hindley (2018), which



has previously been applied to reanalysis data (Wright and Hindley, 2018) and to high-resolution global model data (Lear et al., 2024).

Using this approach, temperature estimates were extracted from the model and used to compute GW $E_p$ and $\lambda_z$, at resolutions

and oversampling volumes consistent with true SABER and GNSS-RO measurements. To derive gravity wave parameters, we then applied the same background removal technique as used for the actual SABER and GNSS-RO temperature data in Section 3.1, followed by applying a second-order Savitzky–Golay low-pass filter with a 3 km frame size to smooth out small-scale variations. This approach is similar to the method described in Hindley et al. (2015). Here we used the Savitzky–Golay filter instead of the weighted sine fitting approach (Section 3.1) because the data length was insufficient for the sine fitting method.

Finally, we applied the S-Transform to the temperature perturbations $T'$ to extract the vertical wavelength $\lambda_z$ and then used Eq. 1 to calculate GW $E_p$.

To understand the differences between the two instruments, we then systematically varied four key parameters representing differences between measurements made by the two real instruments and used the resulting data to assess the impact of these varied parameters on 'measured' GW $E_p$. The four parameters, varied over the ranges described in Table 1, are defined as

follows:

1. Viewing Angle ($\theta$): this is defined as the bearing between the north and the instrument Line-Of-Sight (LOS). For SABER, this angle alternates between 90° off the satellite's along-track travel vector for the northward-looking mode and 270° for the southward-looking mode (Trinh et al., 2015). Real GNSS-RO data have a randomly distributed viewing angle due to their varying orientations.

2. Vertical resolution ($\Delta Z$): This is defined as the FWHM of the vertical averaging implicit in the combined measurement and retrieval of each instrument, and represents the minimum altitude separation at which two atmospheric features can be distinguished in the retrieved profiles.

   – For GNSS-RO, the vertical resolution depends on the bending angle; more information about the determination of vertical resolution of GNSS-RO data can be found in Kursinski et al. (1997).

– For SABER, the vertical resolution is determined by the projection of the field of view (Mertens et al., 2009).

3. Along-LOS resolution ($\Delta X$): This is defined as the FWHM of the instrument's resolution along the Line-Of-Sight (LOS) (Wright and Hindley, 2018), we will refer to it as 'aLOS' in the following text:

   – For GNSS-RO, (Kursinski et al., 1997) defines the aLOS resolution ($\Delta Z$) as the distance traveled by the GPS ray as it enters and exits the atmosphere, approximately 270 km.

– For SABER, aLOS is determined by the instrument's Field-Of-View (FOV), detector size, and the satellite's velocity (Wu et al., 2006; Trinh et al., 2015), and it is on average 400 km (Alexander et al., 2008b).

4. Across-LOS resolution ($\Delta Y$): This is defined as the FWHM of the instrument's resolution across the LOS. This refers to the resolution in the direction perpendicular to the satellite's flight path, affecting the ability to resolve features across



| Parameter | SABER | GNSS-RO | Varying step |
|:---:|:---:|:---:|:---:|
| $\theta$ | 90° | - | 20° |
| $\Delta Z$ | 2 km | 1.2 km | 0.1 km |
| $\Delta X$ | 300 km | 230 km | 10 km |
| $\Delta Y$ | 50 km | 1.5 km | 5.4 km |

**Table 1.** Varied parameters for the model sampled data with the value of each parameter corresponding to each instrument and the varying step. $\theta$: viewing angle, $\Delta Z$: Vertical resolution, $\Delta X$: along-LOS resolution, $\Delta Y$: across-LOS resolution.

the satellite's swath or footprint. This parameter is crucial in remote sensing, as it determines the instrument's capability
to capture features across the satellite's motion. We will refer to this parameter as xLOS.

Table 1 presents the different values selected for each of the four parameters, based on the characteristics of SABER and
GNSS-RO. The viewing angle was varied from $-90°$ to $70°$ in 20-degree increments, with higher angles excluded to prevent
the repetition of the same angles for the selected profiles (Wright and Hindley, 2018). The vertical resolution was varied from
1.2 km to 2 km, in steps of 0.1 km, representing the resolutions of GNSS-RO and SABER, respectively. The aLOS resolution
was varied from 230 km to 310 km in 10 km steps, while the xLOS resolution ranged from 1.2 km to 50 km, with a step size of
5.4 km. We systematically vary each parameter within its defined range and run our sampling code at each specific value. This
process provides the sampled temperature corresponding to each value for each parameter, while keeping the other parameters
fixed as each individual parameter is varied.

## 5.1 Results

### 5.1.1 Gravity Wave Potential Energy

Figure 7 presents the model-sampled GW $E_p$ for SABER (grey) and GNSS-RO (pink). Each panel corresponds to one of the
four studied parameters. The mean GW $E_p$ between 20 and 30 km altitude for each instrument is displayed over the 1-week
simulation period, illustrating how it varies across the range of tested parameter values. The dark-shaded regions represent a
50% confidence interval derived from a bootstrap analysis with 10000 resamplings, while the lighter-shaded areas indicate a
95% confidence interval. It is important to note that the mean shown here is the median of the bootstrap-estimated distribution
of mean states. The results show the following:

1. Viewing angle:

   In general, the GW $E_p$ from the sampled GNSS-RO is roughly twice as high as that from the SABER-sampled GW $E_p$.
   For GNSS-RO, the variation in $E_p$ with respect to the viewing angle is well within our uncertainty range, consistent with
   the true observations being randomly distributed in the viewing angle; specifically, the change in $E_p$ across different
   angles is around 0.5 $\mathrm{Jkg}^{-1}$, indicating a negligible impact. For SABER, during this period, the TIMED satellite faced
   northward. Since it is inclined by $74.1°$ with respect to the true north, the viewing angle should be $90° + 74.1° =$




164.1°. $E_p$ shows a minimum around 10°, with a slight variation of approximately 0.1 Jkg$^{-1}$ between the minimum and maximum values across angles. This suggests that changes in the viewing angle do not significantly affect $E_p$ for either SABER or GNSS-RO.

2. Vertical Resolution:

Both GNSS-RO and SABER exhibit similar behavior in terms of vertical resolution. The GNSS-RO vertical resolution is 1.2 km, while for SABER it is 2 km. The GW $E_p$ in both instruments decreases significantly, compared to other parameters, when transitioning from higher to lower resolution. As vertical resolution improves, the GW $E_p$ increases, capturing a larger portion of the gravity wave spectrum, particularly small-scale waves. The difference in GW $E_p$ between SABER and GNSS-RO is around 2 Jkg$^{-1}$, and this discrepancy is consistent across both instruments. This finding underscores the important role that the vertical resolution plays in estimating GW $E_p$. Additionally, the narrow 50% and 95% confidence intervals for both instruments suggest that the mean GW $E_p$ and its variation with vertical resolution ($\Delta Z$) reliably reflect the impact of resolution on GW $E_p$ values.

3. aLOS Resolution:

The offset between GW $E_p$ from SABER and GNSS-RO remains, with GW $E_p$ from GNSS-RO being approximately twice as large as that from SABER. The impact of aLOS variation on GW $E_p$ is even smaller than the effect of viewing angle. For GNSS-RO, the variation is around 0.2 Jkg$^{-1}$, while for SABER it is less than 0.1 Jkg$^{-1}$.

4. xLOS Resolution:

GNSS-RO exhibits a very slight variation in GW $E_p$ with changes in xLOS resolution. Conversely, SABER shows no variation in GW $E_p$ with changes in $\Delta Y$. The large uncertainty surrounding the mean in both $\Delta X$ and $\Delta Y$ plots indicates that the impact of these parameters on the variability of GW $E_p$ is not significant or clearly defined.

It is important to highlight that these analyses examine only one week of data from a free-running model which is inherently limited in resolution relative to the true atmosphere, albeit high relative to most current global modelling.

The offset of 2 Jkg$^{-1}$ between SABER and GNSS-RO that is seen for all the parameters, except in the vertical resolution, is in agreement with the observations as shown in section 4. This is a good indication that our experiment maintains a geophysically-plausible results.

From the results, it is shown that the vertical resolution plays a major role in the variation of GW $E_p$ for both SABER and GNSS-RO. By increasing the vertical resolution, the GW $E_p$ increased by almost twice the value. This suggests that the offset of the double value of Ep between SABER and GNSS-RO is almost entirely caused by the vertical resolution. We will test this hypothesis in Section 6 using the real observations from both instruments.

### 5.1.2 Vertical Wavelength

Figure 8 shows the distribution of the measured vertical wavelength against each parameter for GNSS-RO (upper panels) and SABER (lower panels), again averaged over the altitudes 20–30 km. The distributions are shown as densities using a normal



kernel density function; probability-density functions have also been tested and show similar results but with much larger noise levels. The ordinate of the kernel density is scaled such that the total area under the density is 1, where the area represents the probability with continuous distributions.

1. GNSS-RO: The distribution of $\lambda_z$ is nearly identical across all four parameters, displaying a bimodal shape with two peaks: one at 6.7 km and another at 8 km. The 8 km peak is slightly more pronounced. This result aligns with the vertical

wavelength distribution presented in Figure 6 of (Hindley et al., 2015), though with slightly higher values in the latter study due to the wavelength selection constraints we applied. The variability among the curves reflects changes in the distribution for each parameter, but this variability is minimal, indicating that the GW $E_p$ remains relatively consistent despite changes in the parameters. The greatest variability between peaks occurs for vertical resolution, with longer $\lambda_z$ becoming more dominant when the vertical resolution is varying.

2. SABER: The distribution of vertical wavelengths for SABER differs from that of GNSS-RO, showing a nearly normal distribution around 8 km across all parameters, except for vertical resolution. The longer dominant vertical wavelength in SABER compared to GNSS-RO aligns with the results from our analysis as well as from previous studies (Wright et al., 2016). The variation in viewing angle also influences the distribution of $\lambda_z$, as anticipated, given the geometry's impact on detecting different wavelengths (Trinh et al., 2015). Changes in viewing angle can cause the satellite's line of

sight to intersect the atmosphere at different altitudes, affecting the observed vertical wavelengths of gravity waves, as noted in studies like (Wu et al., 2006; Ern et al., 2018).

To quantify the variation of $\lambda_z$ with each parameter, we examine the variation of the mean $\lambda_z$ with the variation of the values of each of the four parameters as shown in Figure (9). Again we show the median value of the bootstrap distribution of means for this analysis. The results show that:

– In general, for all the four parameters, SABER displayed longer $\lambda_z$ than GNSS-RO. This agrees with the results shown from the observations as shown in Section 6.

– In SABER, the vertical wavelength slightly changes with the viewing angle. However, this variability is small, on the order of 0.5 km. GNSS-RO on the other hand displayed nearly no change in vertical wavelength with $\theta$. This is reasonable since SABER is observing from a fixed angle while GNSS-RO observes the wave from various angles.

– Vertical resolution is the most important parameter for determining the measured vertical wavelength. For GNSS-RO the relation between the vertical wavelength and the vertical resolution is almost linear, i.e., the coarser the vertical resolution, the longer the vertical wavelength detected by the instrument. This is not the case for SABER, where varying the vertical resolution does not strongly impact the measured vertical wavelength. This assures the previous results that shown in Figure 8(a), (b), and (c) that a combination of the three parameters most probably influences the vertical

wavelength detected by SABER; $\theta, \Delta Z$, and $\Delta X$.





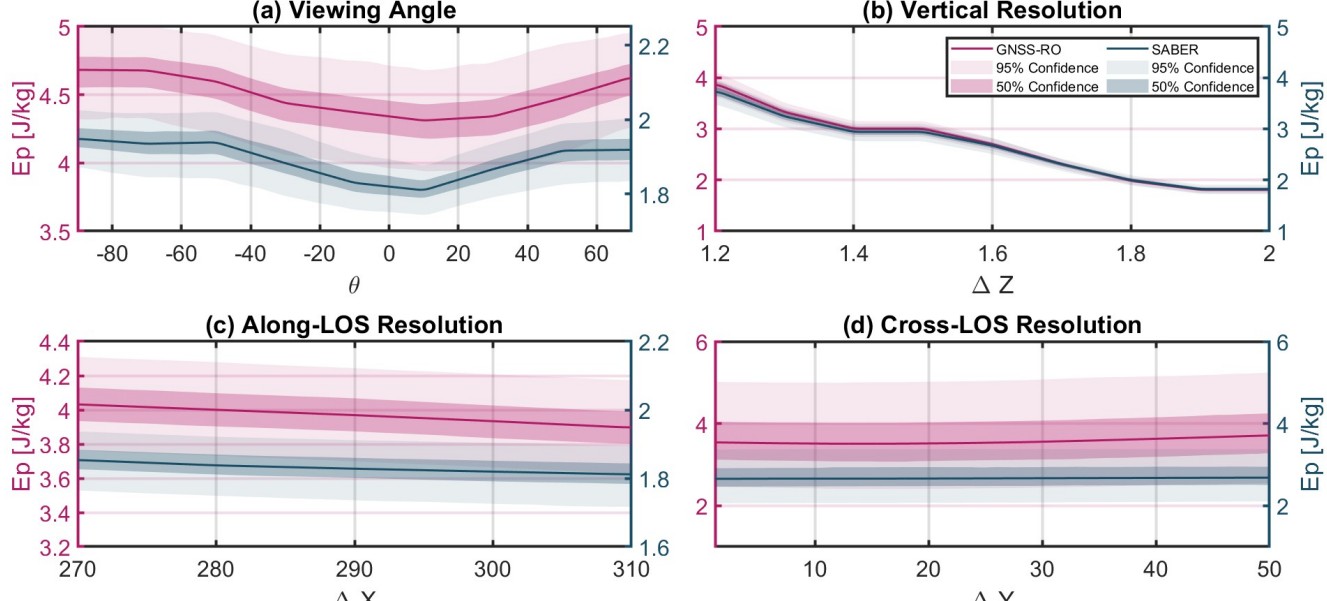

**Figure 7.** The variation of the model-sampled bootstrap mean of GW $E_p$ with respect to the four parameters: $\theta$ (a), $\Delta Z$ (b), $\Delta X$ (c), and $\Delta Y$ (d). The darker shading represents the 50% and the lighter shading indicates the 95% confidence intervals around the mean.

- aLOS and xLOS resolutions have almost no effect on the vertical wavelength, particularly for GNSS-RO. For SABER, aLOS has a slight influence on vertical wavelength detection. Since the wave fronts in the model are inclined relative to the instrument's horizontal and vertical detection axes, the horizontal sampling distance can influence the estimation of vertical wavelength. For instruments with a longer horizontal sampling range, the averaging differs slightly compared to those with shorter ranges, introducing aliasing. This causes a tendency toward detecting longer $\lambda_z$ with greater horizontal sampling distances, as seen with SABER, and shorter $\lambda_z$ with shorter horizontal distances, as with GNSS-RO.

## 6  Adjusting Observations

As demonstrated above, the difference in vertical resolution between SABER and GNSS-RO is the primary factor responsible for the discrepancies in the GW properties they measure in the QBO-dominated tropical lower stratosphere, particularly when considering GW $E_p$. In this section, we build on this finding to explore how the observations can be considered in a more equivalent way for future work.

We analyzed the GEOS model temperatures sampled as GNSS-RO and SABER retrieved temperatures. Our results identified the vertical resolution as the most crucial factor contributing to the differences in $E_p$ and $\lambda_z$. Based on these results, we then adjusted the GNSS-RO observations to match SABER's vertical resolution. We apply a boxcar smoothing to the raw GNSS-RO temperature profiles with an interval of 2 km in altitude before performing the GW analysis. Initially, the GNSS-RO temperature





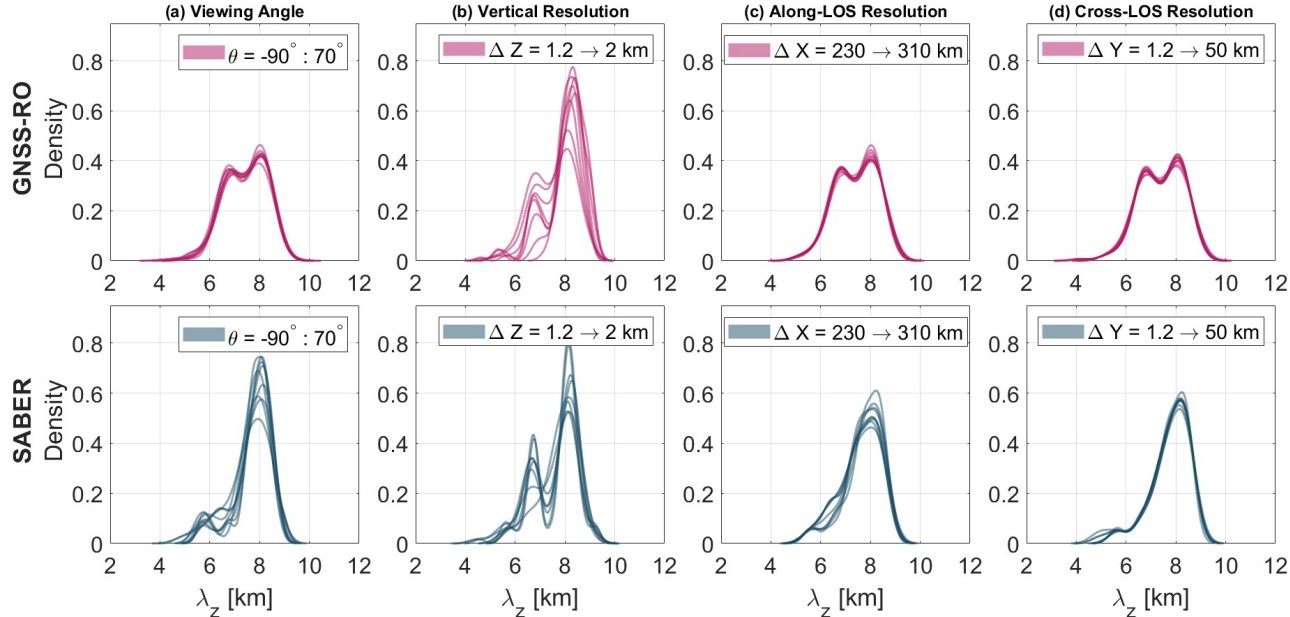

**Figure 8.** Kernel density of $\lambda_z$ distributions at different parameters; $\theta$ (a), $\Delta Z$ (b), $\Delta X$ (c), and $\Delta Y$ (d) for GNSS-RO (upper row) and SABER (lower row). The legend indicates the range of the values for each of the four parameters.

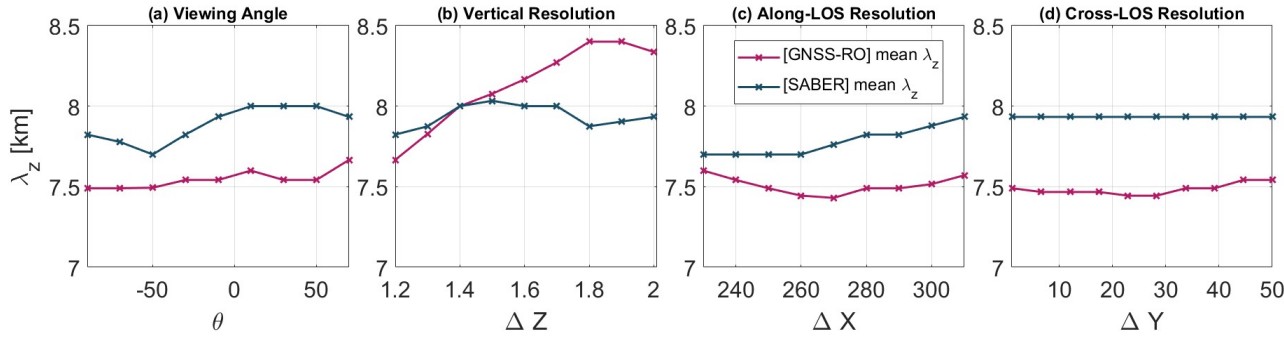

**Figure 9.** Median $\lambda_z$ varied with four parameters; $\theta$ (a), $\Delta Z$ (b), $\Delta X$ (c), and $\Delta Y$ (d) shown for GNSS-RO (pink) and SABER (blue).

data had a resolution of 1.2 km. We then applied the method described in Section 3.1 to the smoothed temperatures to extract gravity wave properties.

Next, we recreated Figures 5 and 6 using the $E_p$ estimation from the smoothed GNSS-RO temperatures for a given regular QBO cycle. Figures 10 and 11 show the resulting absolute differences in GW $E_p$ and $\lambda_z$, respectively, with the left panels showing results before and the right panels after smoothing GNSS-RO temperatures.





## 6.1 GW $E_p$

After smoothing, the largest differences are observed in regions where the GNSS-RO GW $E_p$ was initially much higher than that of SABER, particularly in areas where the Ep difference exceeded 5 $\mathrm{Jkg}^{-1}$ (Fig. 10(a)). This mainly occurs within the eastward wind shear during the QBO's eastward phase at lower altitudes and at the transition from the EQBO to the WQBO phase. Post-smoothing, these discrepancies are significantly reduced to approximately $\pm 1$ $\mathrm{Jkg}^{-1}$, consistent across all altitudes and during all QBO phase transitions.

## 6.2 Vertical Wavelength $\lambda_z$

The impact of smoothing on vertical wavelength differs from its effect on $E_p$. The difference in vertical wavelengths decreases at some altitudes and reverses at others. Before smoothing, GNSS-RO consistently detected shorter vertical wavelengths than SABER. After smoothing, GNSS-RO wavelengths become longer than those of SABER below approximately 22 km and above around 29 km. Between 22 and 29 km, the vertical wavelengths from both instruments are nearly identical, with GNSS-RO showing slightly shorter wavelengths than SABER. This is consistent with the model-sampled data discussed in Section 5.1.2, which showed that the relationship between $\lambda_z$ and vertical resolution for SABER is not linear, suggesting the influence of additional factors.

These findings, along with the model data, suggest that factors beyond vertical resolution, such as a variation or a combination of variation viewing angle, and vertical and aLOS resolutions, are contributing to the change in vertical wavelength detection. To quantify this combined effect, we normalized the vertical wavelength variations with respect to each of the three parameters—$\theta$, $\Delta Z$, and $\Delta X$—to a range between 0 and 1. We then calculated the mean $\lambda_z$ variation for each parameter and summed these values to represent the total variation in $\lambda_z$. Finally, we determined the contribution of each parameter to the total variation by calculating the ratio of the individual means to the total. The results showed that $\theta$ accounts for 32% of the total variation, $\Delta Z$ contributes 36%, and $\Delta X$ contributes 32%.

## 7 Summary and Discussion

There are several key points in this study that we believe it is important to highlight and discuss. First, the periodogram results in Section 4.1 showed that $E_p$ from SABER remains stable regardless of the study period length, consistently revealing a persistent annual cycle in the time series. This annual cycle becomes more pronounced with a longer period of study, which aligns with previous findings. For instance, John and Kumar (2012) showed a seasonal variation of $E_p$ from SABER in the tropics, superimposed by the same periodicity of the regular QBO, and also confirmed earlier findings by (Ern et al., 2014) that the QBO modulates GW temperature amplitudes.

However, the variability in $E_p$ observed from GNSS-RO depending on the length of the study period suggests a stronger modulation by the QBO. The dominant fluctuations in $E_p$ from GNSS-RO coincide with QBO periods, particularly the disruption periods, implying that the QBO more strongly influences the gravity wave spectra observed by GNSS-RO. This agrees





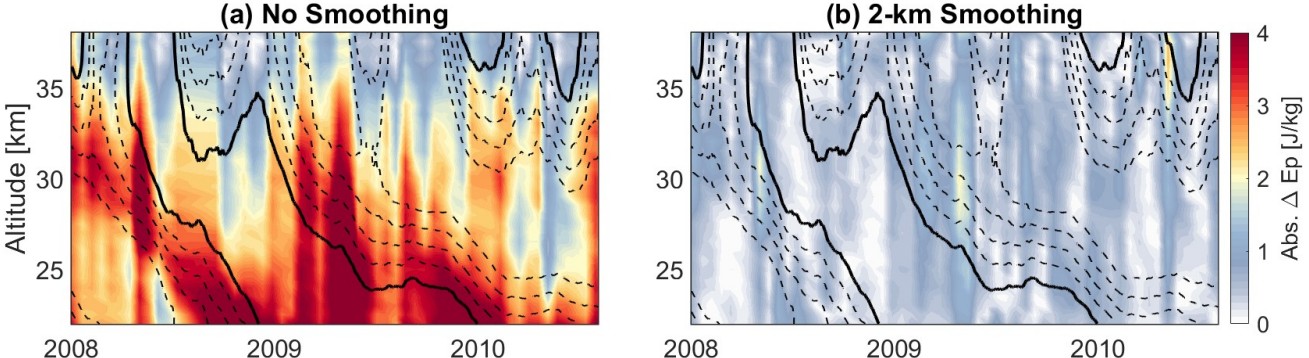

**Figure 10.** Time-height cross section of the absolute difference in $E_p$ between GNSS-RO and SABER during an arbitrary regular QBO cycle as shown by the color map, overlaid by the contour lines of the zonal winds. solid lines are the zero-wind line and dashed lines are westward winds with a spacing of 5 $ms^{-1}$. Panel (a) is showing the difference in $E_p$ before smoothing and panel (b) is the difference after the 2 km smoothing.

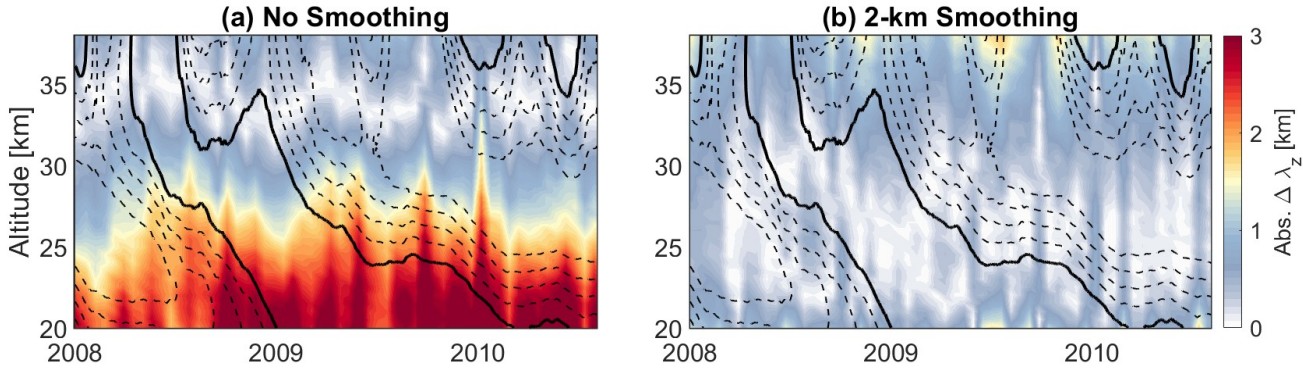

**Figure 11.** Same as Figure 10 but for $\lambda_z$.

with earlier studies showing a pronounced QBO signal in GNSS-RO data (e.g., Namboothiri et al., 2008; Tsuda et al., 2009; Nath et al., 2014). Further investigation into the interaction between gravity waves and the QBO during QBO disruptions is necessary and could be addressed in future work.

Second, the comparison of $E_p$ between SABER and GNSS-RO, as illustrated in Fig. 5, indicates that GNSS-RO consistently observes higher GW $E_p$ than SABER, particularly at lower altitudes. This is due to GNSS-RO's stronger sensitivity to shorter spatial scales and lower altitudes, allowing it to capture a wider part of the gravity wave spectrum than SABER. This effect is evident in Fig. 6, where GNSS-RO exhibits shorter vertical wavelengths compared to SABER.

    This outcome aligns with previous studies. For instance, (Tsuda et al., 2009) found that GPS RO data consistently showed

maximum $E_p$ in the tropics within the 19–26 km altitude range across boreal winter and spring. Similarly, (Faber et al., 2013)



identified high $E_p$ from GPS RO in the tropics within 20–30 km, corresponding to shorter vertical wavelengths. However, (Liu et al., 2017) reported that maximum $E_p$ derived from SABER was observed at higher latitudes rather than in the tropics. Although our analysis of SABER and GNSS-RO $E_p$ included only vertical wavelengths shorter than 12 km, differences between the results from each instrument are still evident. This difference may arise from the strength of zonal winds and the dominant

gravity wave sources in the tropics compared to high latitudes. In higher latitudes during winter, stronger zonal winds tend to refract waves, elongating their wavelengths, which makes them more detectable by SABER. In contrast, the primary source of gravity waves in the tropics is convection; these convective gravity waves have a high frequency and are therefore more easily captured by GNSS-RO. Highlighting these differences is essential, as it underscores the specific suitability of each instrument for different types of gravity wave studies.

To pinpoint the observational filter differences between SABER and GNSS-RO, we used 3 km resolution GEOS temperatures to generate synthetic retrieved temperatures for both instruments as shown in Section 5. This model choice is well-suited for the study due to its high vertical resolution and accurate representation of the observed QBO (Ho et al., 2023). (Wright and Hindley, 2018) demonstrated that the sampling method employed in this study showed a high correlation between sampled temperatures and actual SABER and GNSS-RO observations within the altitude range of 20–40 km. The sampled $E_p$ revealed

noteworthy results. The consistent offset observed between SABER and GNSS-RO across all observational filter parameters, except the vertical resolution, is mainly compensated by differences in actual vertical resolution between the two instruments; however, this is less straightforward for vertical wavelength, especially in SABER data. Figures 8 and 9 indicate that $\lambda_z$ from SABER results from a combination of these three parameters, rather than solely vertical resolution as in GNSS-RO. Preliminary estimates indicate that each parameter contributes nearly equally to the variability in vertical wavelength.

We used the results from the model-sampled data to account for observational filter differences between the true measurements by reducing the vertical resolution of GNSS-RO temperatures to match SABER's temperature profiles. Figures 10 and 11 illustrate the reduced discrepancies between the true observations after smoothing the GNSS-RO data, demonstrating that the model can accurately replicate the observations and confirming that vertical resolution is the main driver of the observational filter differences between the SABER and GNSS-RO. such a result suggests that GNSS-RO data can potentially extend

the study of GW-QBO interactions, which has been conducted with SABER for over 23 years. This outcome is crucial for establishing consistency between both instruments and for advancing GW-QBO research with the future GNSS-RO missions.

    This study complements previous work by:

    – Focusing on the tropical region, where the QBO occurs, to analyze GW-QBO interactions specifically from each instrument. Prior studies comparing SABER and GNSS-RO have either targeted other regions (e.g., Wright et al., 2016) or

595        conducted global analyses (e.g., Wright et al., 2011).

    – Quantifying and identifying the differences in observational filters between SABER and GNSS-RO through sampled model data. Previous studies have used sampled data either to compare with model outputs, such as (Lear et al., 2024) or to assess how well reanalysis can represent satellite-observed gravity waves (Wright and Hindley, 2018).



Additionally, this study provides a direct comparison between the most widely used satellites for GW-QBO interaction studies: SABER, as demonstrated in (e.g., Ern et al., 2014), and GNSS-RO, used in (Tsuda et al., 2009; de la Torre and Alexander, 2005; Nath et al., 2014). It presents a one-to-one comparison of $E_p$ and $\lambda_z$ observed by both instruments and quantifies each instrument's limitations in capturing QBO-modulated gravity waves.

## 8 Conclusions

Our study confirms previous findings on the differences in observational filters between SABER and GNSS-RO when observing GW-QBO interactions, even when analyzing within the same vertical wavelength range and applying the same analysis methods. This work highlights how each instrument captures specific portions of the gravity wave spectrum, emphasizing the importance of selecting an appropriate instrument for targeted gravity wave studies. By quantifying these observational differences using model data, we demonstrate how effectively high-resolution model data aligns with actual observations. This approach also provides insights into the limitations of each observation method and helps address hypothetical questions about observational mechanisms.

The significance of our study also lies in demonstrating that GNSS-RO can extend the long-term research on gravity wave modulation by the QBO, which SABER has conducted for over 23 years. This result is particularly valuable, as GNSS-RO will continue with new missions, while SABER nears the end of its operational life.

*Data availability.* The repository of GNSS-RO data in the AWS Registry of Open Data can be found at http://registry.opendata.aws/gnss-ro-opendata/. Documentation of the repository, source code for the API, and tutorial demonstrations are published under a digital object identifier (Leroy and McVey, 2023). The database API is published through PyPI; see http://pypi.org/project/awsgnssroutils/. The SABER/TIMED data used in this work can be obtained from https://saber.gats-inc.com/. DYAMONDv2 data can be accessed via the NCCS Dataportal - Datashare (https://portal.nccs.nasa.gov/datashare/G5NR/DYAMONDv2/).

*Author contributions.* The conceptualization of the study, software, and validation of the results were done by M. Almowafy, C. Wright, and N. Hindley. Formal analysis was done by M. Almowafy and C. Wright. Investigation, visualization, and writing the original draft is done by M. Almowafy. Data curation, funding acquisition, and project administration was done by C. Wright. Methodology and supervision were done by C. Wright and N. Hindley. All authors contributed to the interpretation of the results and the final text.

*Competing interests.* The authors declare that they have no conflict of interest.



*Acknowledgements.* This study was supported by the UK Natural Environment Research Council (NERC) under grant numbers NE/W003201/1

supporting M. Almowafy, C. Wright and N. Hindley, and NE/V01837X/1 supporting C. Wright. N. Hindley was also supported by a NERC

Independent Research Fellowship NE/X017842/1, and C. Wright was also supported a Royal Society University Research Fellowship UR-

F/R/221023. M. Almowafy and C. Wright were additionally supported by Royal Society grant no. RF/ERE/221011. DYAMOND data man-

agement was provided by the German Climate Computing Center (DKRZ) and supported through the projects ESiWACE and ESiWACE2.

The projects ESiWACE and ESiWACE2 have received funding from the European Union's Horizon 2020 research and innovation programme

under grant agreements No 675191 and 823988. This work used resources of the Deutsches Klimarechenzentrum (DKRZ) granted by its

Scientific Steering Committee (WLA) under project IDs bk1040 and bb1153. The ChatGPT LLM was used for minor refinements of textual

style at the individual-sentence level, but all scientific results were obtained using non-AI methods.



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
