# Peer review of "Stratospheric gravity waves in a post-limb sounder era: can GNSS-RO be used to extend the SABER QBO-driving record?"

_EGUsphere, 2024_

## Referee Comment (RC2)

Review of **Observations of stratospheric gravity waves in the tropics: can GNSS-RO extend the SABER climatological record?** by Almowafy et al.

This paper provides an interesting and in-depth analysis of observational methods for gravity wave research via the SABER instrument and the GNSS-RO technique. The authors provide a well-written and thorough manuscript describing their methods and the important takeaways from their work. The focus on directly comparing SABER with GNSS-RO observations and on the implications of observational filters with the purposes of continuing the climatological record of gravity waves after SABER is decommissioned is novel and unique and will be of interest to the AMT audience. However, there are a few key details that should be considered before the manuscript is published. My in-depth comments, primarily minor in nature, are below.

General Comments:
1.  Section 2.2: With different instruments and centers using different processing techniques of the 13 different GNSS satellites/missions, could this have an impact on your results? For example, is there any sensitivity or difference between your results if only COSMIC were used vs. only Spire? Also, what is the breakdown of profiles from each of the different missions? This would also help to demonstrate how different this study is from the COSMIC/SABER work in Wright et al. 2011 & 2016.
2.  This is a bit more of a technical correction but there are some issues with citations throughout, my guess is just from a mix-up of \cite, \citet, and \citep in LaTeX. I've caught a few of them and written their line numbers here, but suggest double checking this throughout the manuscript:
    a.  Line 181, 183, 203, 204, 382, 413, 564, 565, 566/567, 577/578, 597, 600, 601
3.  Section 4.1 and Figure 4: This section would greatly benefit with some statistical significance testing of the power spectra peaks. I especially struggle with Line 315-316 and the statement that there is a persistent peak at 1.25 years evident in all three datasets, where the zonal wind peak is very small (and smaller than other, seemingly random, peaks). Significance testing against a red noise hypothesis and a 95% significance curve would do a lot to strengthen your argument; or if the results end up being not statistically significant, could help inform your analysis and discussion.
4.  In the current discussion and conclusions section, it feels that the use of GNSS-RO to extend SABER is more of an afterthought rather than the purpose of this paper based on the title. I strongly recommend adding more detail here to make the story more compelling. One potential suggestion would be to add a figure showing a comparison of the time series of Ep across the period of SABER and the smoothed GNSS-RO data at some height or for some layer. If the time series line up well, this would be a good demonstration of the ability for RO to continue the GW climatology of SABER.

Specific Comments:

1. Line 32: Please define SABER and GNSS-RO at first use
2. Lines 61-62: The results of the previous study mentioned here seem particularly relevant to this work and should be mentioned/discussed further here. Additionally, please define COSMIC at first use.
3. Line 68: I would be careful at calling GNSS-RO an "instrument", perhaps calling it a "technique" here is more fitting?
4. Line 77-79: use the main ERA5 citation at the end of this first sentence
5. Line 85: What is the vertical resolution in the stratosphere?
6. Section 2.2: Mention the GNSS vertical resolution here, like how the SABER resolution is mentioned in 2.3
7. Section 3.1: I'm a bit confused as to why the GNSS-RO data is oversampled and interpolated to 0.1 km levels (more than 10x the original resolution) while the SABER data is interpolated to 0.5 km (4x the original resolution) to avoid oversampling. Then the RO data is downsampled to 0.5 km anyway, so what is the purpose of the intermediate step? Please explain.
8. Line 130/575 and throughout: Please ensure proper use of "grid spacing" rather than "grid resolution" where appropriate
9. Lines 176-178: The ending of this paragraph feels a little jumbled and a bit unfinished, specifically in the last sentence. I suggest reworking to make clear that the RO dataset coverage is not sufficient for the method being described.
10. Line 180: "...significant advantages in terms of more robust derived values over a binning approach" → what does this mean exactly?
11. Figure 2: I suggest either reorganizing this figure or adding in additional titles/labels to make the differences in the panels clearer. Specifically, since there is no x-label for panel b, it looks as if the panel has a shared x-axis range of 28 months as panel c. Additionally, add a label to the colorbar for panel c
12. Line 217-218: This is a pretty subjective statement (though I don't disagree), maybe add a little more detail about their main differences?
13. Line 230-231: I had to read this a couple of times to understand, I suggest reordering the sentence for clarity so that it reads "these two metrics are chosen as they can be measured easily using both instruments, unlike more heavily-derived..."
14. Line 284: where does this "90% larger" number come from? Should this say 900%? Or 9 times larger?
15. Line 485-486/Figure 4: I think it is important to mention that the SABER distributions do have a more pronounced tail to smaller wavelengths.
16. Figure 10 caption: I think this caption needs to be a little more descriptive (particularly in the last line) so that it explains what the 2 km smoothing is applied to even though it is described well in the text
17. Line 571-573: Please add a citation here. Additionally, it could be helpful to provide approximate wavelengths for convection
18. Line 579/580: "The sampled Ep revealed noteworthy results" is a bit of an abrupt transition, I suggest reworking

Technical Corrections:

1. Line 47: "that SABER instrument" → "that the SABER instrument"
2. Line 182: PWs → GWs
3. Line 215: include → includes
4. Line 374: "the observational filters of these" → "the observational filters are of these" or something similar
5. Line 589: such → Such

---

## Author Response (AR1)

**Response to Reviewers**

We thank the reviewers for their comments and the time spent preparing them.

**Reviewer 1**

**Major comments:**

1. The title: "can GNSS-RO extend the SABER climatological record?". The main content of this work is the effects of observational filters on the parameters of derived GWs from SABER and GNSS-RO. The title does not match the main content. Please clarify the extent and/or the aspect on which the GNSS-RO can extend the SABER climatological record.

The main purpose of this paper is not to just address the observational filter differences between GNSS-RO and SABER measurements, but also to show that given the known observational filter differences between both techniques. We find that by simply smoothing GNSS-RO raw data with 2km we can largely reproduce SABER results at the tropics, which is important in the sense that SABER is in its 23rd year of an original 2-year mission and we want to make sure that the data and the results from this valuable observation can be continued. GNSS-RO on the other hand is launching new missions and will continue to provide measurements in the future. This main purpose has been highlighted in the Abstract and in the conclusions with an additional figure (Fig. 12 ) that is added to show the matching timeseries of both GNSS-RO and SABER over the studied period (2007 – 2022) at the tropics. Therefore, we believe that the title now better matches the body of the paper and hence is suitable for the revised manuscript.

2. Abstract: Only the purpose and method of this work are presented. It is better to present some quantitative results in the abstract.

This has added to the revised version of the paper (L16 – L19).

- 3. Comparing the difference of Ep and vertical wavelength: the difference between those derived from SABER and GNSS-RO are compared in the manuscript. However, the absolute of values of Ep and vertical wavelength derived SABER or GNSS-RO are not known, consequently, the statistical significance of the difference is unknow. Otherwise, one may provide the percentage contributions of Ep (and vertical wavelength) on the total Ep from both SABER and GNSS-RO. Also, the statistical significance of the differences should be provided.
- 2 Figures have been added to the Appendix section under supplementary figures to show the absolute values of Ep and vertical wavelength (Fig. A1 and Fig. A2, respectively) for each instrument, and the reader is referred to them in L352.
- 4. Please summarize the main similarities and differences of SABER and GNSS-RO in studying the GW-QBO interactions.

A summary of the main similarities and differences between GNSS-RO and SABER is added to section summary and discussion starting from L582 to L595.

**Minor comments:**

1. Figure 2 caption, "the thick"-->"The tick". The ticks are unreadable in panel (c) of Figure 2 and in other contour figures (Figures 3, 5, 6, 10, 11).

Done for all the mentioned figures as well as the supplementary figures.

2. L266: How to determine the upper limit of horizontal wavelength (2000 km) measured by SABER and GNSS-RO?

It is not possible to determine the upper limit of horizontal wavelength measured by GNSS-RO, it is only an estimate. The upper value is omitted to avoid confusion.

3. L269: Since the lower limits of vertical wavelengths are different (2 km for GNSS-RO, 4 km for SABER). For a fair comparison, the analysis to GWs to GWs with vertical wavelength shorter than 12 km. The lower limits of the vertical wavelengths of both measurements should also be mentioned.

We did not fully understand the first part of the comment, however the lower limits of both GNSS-RO and SABER data used in the study are already mentioned for GNSS-RO: in L265 and for SABER: in L268.

- 4. L278-279: "panel (a)"-->"panel (a) of Figure 3"? "panel (b)"-->"panel (b) of Figure 3"? Fixed.
- 5. Section 4.2: There is no description on the 2019 disruption.

A description of the 2019 disruption is added to the revised version of the paper (L371- L373).

- 6. L366: Where can I refer and how can I understand the statement "the shorter wavelengths detected by GNSS-RO are associated with a larger range of Ep compared to SABER"? Omitted for brevity.
- 7. Conclusion: It is better to conclude the new finding(s) and/or limitations more specifically and quantitatively of this work. Other than the qualitative statements such as, "emphasizing the importance of selecting an appropriate instrument for targeted gravity wave studies", etc.

The conclusion section is rephrased in the revised version to highlight the importance and the main purpose of the study.

**Reviewer 2**

**General Comments:**

1. Section 2.2: With different instruments and centers using different processing techniques of the 13 different GNSS satellites/missions, could this have an impact on your results? For example, is there any sensitivity or different between your results if only COSMIC were used vs. only Spire? Also, what is the breakdown of profiles from each of the different missions? This would also help to demonstrate how different this study is from the COSMIC/SABER work in Wright et al. 2011 & 2016.

A new figure is added to the revised manuscript (Fig. 13) in section 6.3 which shows how Ep varies between the different instruments if they were selected individually and how the density of the data coverage increases with time to improve the accuracy of Ep estimation. Text has been added to explain the impact of the number of profiles per day on the estimation of Ep (L593-L604). More text has also been added about why this study is different from Wright et al. 2011 & 2016 in the introduction section (L66-L70)

- 2. This is a bit more of a technical correction but there are some issues with citations throughout, my guess is just from a mix-up of \cite, \citet, and \citep in LaTeX. I've caught a few of them and written their line numbers here, but suggest double checking this throughout the manuscript:

  a. Line 181, 183, 203, 204, 382, 413, 564, 565, 566/567, 577/578, 597, 600, 601

  Fixed. Thank you very much for providing the line numbers of the citations that require correction.
- 3. Section 4.1 and Figure 4: This section would greatly benefit with some statistical significance testing of the power spectra peaks. I especially struggle with Line 315-316 and the statement that there is a persistent peak at 1.25 years evident in all three datasets, where the zonal wind peak is very small (and smaller than other, seemingly random, peaks). Significance testing against a red noise hypothesis and a 95% significance curve would do a lot to strengthen your argument; or if the results end up being not statistically significant, could help inform your analysis and discussion.

Statistical significance is calculated for the primary peaks of all of the three time series; zonal wind, GNSS-RO Ep and SABER Ep using False Alarm Probability and added to the revised manuscript in L334-L339. Thank you for the suggestion, it added more reliability to the results.

4. In the current discussion and conclusions section, it feels that the use of GNSS-RO to extend SABER is more of an afterthought rather than the purpose of this paper based on the title. I strongly recommend adding more detail here to make the story more compelling. One potential suggestion would be to add a figure showing a comparison of the time series of Ep across the period of SABER and the smoothed GNSS-RO data at some height or for some layer. If the time series line up well, this would be a good demonstration of the ability for RO to continue the GW climatology of SABER.

Thank you very much for this suggestion. A new figure (Fig. 12) and a new subsection (section 6.3) are added to the revised manuscript in the section on adjusting the observations showing a comparison of the time series of Ep across the period of SABER and the smoothed GNSS-RO data at  $Z=30\,\mathrm{km}$ . The matching between both time series confirms the main purpose of the paper to use GNSS-RO to extend SABER. The figure also shows the distribution of the Ep from both instruments at all heights and how the smoothing of GNSS-RO brought the distribution of its data closer to SABER Ep distribution.

**Specific Comments:**

- **1. Line 32: Please define SABER and GNSS-RO at first use** Fixed.
- 2. Lines 61-62: The results of the previous study mentioned here seem particularly relevant to this work and should be mentioned/discussed further here. Additionally, please define COSMIC at first use.

See response 1 in the general comments. The second part is fixed.

3. Line 68: I would be careful at calling GNSS-RO an "instrument", perhaps calling it a "technique" here is more fitting?

Done.

- **4.** Line 77-79: use the main ERA5 citation at the end of this first sentence Done.
- 5. Line 85: What is the vertical resolution in the stratosphere?

This has been added to the revised text (L93).

6. Section 2.2: Mention the GNSS vertical resolution here, like how the SABER resolution is mentioned in 2.3

Done (L111-L112).

7. Section 3.1: I'm a bit confused as to why the GNSS-RO data is oversampled and interpolated to 0.1 km levels (more than 10x the original resolution) while the SABER data is interpolated to 0.5 km (4x the original resolution) to avoid oversampling. Then the RO data is down sampled to 0.5 km anyway, so what is the purpose of the intermediate step? Please explain.

This is mainly for file storage reason and does not affect the results. we store it at this resolution for other purposes.

8. Line 130/575 and throughout: Please ensure proper use of "grid spacing" rather than "grid resolution" where appropriate.

Fixed.

9. Lines 176-178: The ending of this paragraph feels a little jumbled and a bit unfinished, specifically in the last sentence. I suggest reworking to make clear that the RO dataset coverage is not sufficient for the method being described.

The paragraph is rephrased and additional information about the method is added to make it clear (L184-L188).

10. Line 180: "...significant advantages in terms of more robust derived values over a binning approach" what does this mean exactly?

This has been rewritten in a better way to clearly understand what it means. (L190)

11. Figure 2: I suggest either reorganizing this figure or adding in additional titles/labels to make the differences in the panels clearer. Specifically, since there is no x-label for panel b, it looks as if the panel has a shared x-axis range of 28 months as panel c. Additionally, add a label to the colorbar for panel c

The x-axis labelling is moved to the top to make it more clear to the reader. A label is added to the colorbar for panel c.

12. Line 217-218: This is a pretty subjective statement (though I don't disagree), maybe add a little more detail about their main differences?

Fixed and more details has been added to the revised version (L228-230)

13. Line 230-231: I had to read this a couple of times to understand, I suggest reordering the sentence for clarity so that it reads "these two metrics are chosen as they can be measured easily using both instruments, unlike more heavily-derived..."

Fixed (L243-L245).

14. Line 284: where does this "90% larger" number come from? Should this say 900%? Or 9 times larger?

Fixed (L297-L298).

15. Line 485-486/Figure 4: I think it is important to mention that the SABER distributions do have a more pronounced tail to smaller wavelengths.

Added (L511-L512).

16. Figure 10 caption: I think this caption needs to be a little more descriptive (particularly in the last line) so that it explains what the 2 km smoothing is applied to even though it is described well in the text

Done.

17. Line 571-573: Please add a citation here. Additionally, it could be helpful to provide approximate wavelengths for convection

Added (L644-645).

18. Line 579/580: "The sampled Ep revealed noteworthy results" is a bit of an abrupt transition, I suggest reworking

Rephrased (L651-L652).

**Technical Corrections:**

1. Line 47: "that SABER instrument" -> "that the SABER instrument"

Fixed.

2. Line 182: PWs → GWs

Fixed.

3. Line 215: include > includes

Fixed.

4. Line 374: "the observational filters of these"  $\rightarrow$  "the observational filters are of these" or something similar

Fixed.

5. Line 589: such  $\rightarrow$  Such.

Fixed.

---

## Author Response (AR2)

**Response to Reviewers**

We thank the reviewers for their comments and the time spent preparing them.

**Reviewer 1**

**Major comments:**

1. L9-13: "To test this ... the same results": These sentences show your purpose and main procedures. It is better to provide your main conclusions here. such as, how about the observed GWs depend on the viewing angles, the vertical and horizontal resolutions of SABER and GNSS-RO by using the same atmosphere data (high-resolution GEOS model). However, in the subsequent sentences, I think the GW Ep were calculated from the actual observations from SABER and GNSS-RO.

The GW Ep is calculated twice, once from observations and another from model for the purpose of comparison. However, there is no possibility to measure how the observed GW Ep would change with the angle or the resolution as we cannot change the instrument itself, that is the reason why we used the model data and sample it as similar as possible to the observations. We could then alter the viewing angle and the resolutions hypothetically and see how the results help us with the observations. The model results show that the vertical resolution is the most effective factor out of the 4 ones tested. We applied this result back to the observations and proved that it is correct.

**2. L14 and L19: Does the 1.3 Jkg-1 and 0.5 Jkg-1 are different from zero in statistical sense? These two distributions are stat. sig.**

We have carried out a test of statistical significance using the non-parametric two-sample Kolmogorov-Smirnoff test, which we describe the results of in the section on 'Time series and Distribution', and have mentioned in the abstract that these values are statistically significant. The choice of a non-parametric test is due to the non-Gaussian nature of the SABER and smoothed GNSS-RO potential energy distributions, as shown in Figure 12b.

3. L15-17: "...the most significant factor in determining...", How about the quotative relation between the difference (between the results of both instruments) between the vertical resolution? Does the resolution mean the vertical step of between two consecutive sampling points. Yes, vertical resolution and sampling point spacing are different – the vertical step between sampling points is the spatial separation between the centres of two infrared (SABER) / radio (GNSS) point spread functions which define the spatial lower bound of features that can be distinguished, i.e. the PSF represents the true resolution, which in both cases is much coarser than the step spacing. This is discussed in section 5 (L435), where we define what the vertical resolution is and how it varies from one instrument to another depending on how each instrument is designed.

**4. The second paragraph of the Abstract: What is the conclusion of this paragraph in supporting the "interactions between GWs and the QBO" as stated in L6-9**

Our ability to **observationally** constrain the physics of interaction between GWs and the QBO is inherently and fundamentally limited by our ability to accurately measure these processes. While the bulk QBO is fairly easy to constrain accurately even in reanalysis, this is much more of a technical challenge for GWs. By assessing the effect of the observational filter on the GW-QBO interaction, we are able to draw conclusions about our fundamental ability to say anything about these interactions at all.

Our study and the conclusions we draw quantify how much we can trust these measurements in the context of this observational filter differences between the two sets of measurements, feeding back up the chain to the primary question of measurement reliability and consistency for long term study.

5. The third paragraph of the Abstract: This paragraph only shows the essential rule of studying gravity waves. The question is what are the appropriate observational methods for gravity wave research of this work?

This paragraph is to emphasis that GNSS-RO is still capturing short vertical wavelengths of gravity waves than SABER, but when we adjust the vertical resolution of GNSS-RO to match that of SABER, we get very similar results, hence we can use GNSS-RO to extend the SABER climatology. This is demonstrated in detail in the results section of the paper.

6. Please clarify the terms used in the text: resolution, sampling step, lower limit of the vertical wavelength used to derived Ep.

The term resolution is defined in section 5 (L435). The term sampling step is defined in section 3.1 L171 – 175. The lower limit of vertical wavelength for each instrument is defined in section 4 L281 -285.

7. Looking through the abstract, I still cannot get the main purpose of this manuscript. In the response: As stated in the response, the main purposes are: (1) address the observational filter differences between GNSS-RO and SABER measurements, (2) given the known observational filter differences between both techniques, (3) validate the reliability of SABER observation through comparing the GW derived from both SABER and GNSS-RO. Please fulfil these purposes in the abstract since the abstract should not provide only the purpose of the work but provide a concise summary of the research purpose, methods, main results, and conclusions. After these issues are resolved, you may find a proper title.

The main purpose of this manuscript is (1) and (2): given the known observational filter differences between both techniques, we identify the main reason behind the difference in GW Ep between both instruments. This is confirmed because, when we adjust GNSS-RO to match SABER's vertical resolution, we get good agreement between both timeseries. We have already discussed in our previous response how the title corresponds to the main results of the paper and to strengthen this title-to-content linkage, we have added more material including Fig. 12 and additional text in the Abstract, summary and conclusions that complement the title and answer the question in it. Due to these changes and in the context of our previous arguments, we believe that the title is descriptive of the work presented but are happy to reconsider this on request of the editor.

**Minor comments:**

1. L100: Please provide some references of "other datasets".

This was a misleading sentence; the choice of temporal and spatial coverage is to ensure consistency with SABER and GNSS-RO. This is fixed in the revised version 2.

2. L118: What is the mean of "vertical resolution of ~1 km". It is better to say that the temperature profiles have a uncertainty or accuracy of? K with vertical sampling step of 1 km in the stratosphere. Corresponding revisions should be made for SABER temperature (L132-133). Otherwise, if one does not consider the temperature uncertainty or accuracy, the vertical sampling stem can be arbitrary through over-sampling or down-sampling or interpolation.

Fixed.

3. L122-125: How about the consistency of the GNSS-RO measurements from different missions? i.e., the standard derivations of the concurrent temperature profiles measured from different missions.

Fig. 13 was added based on a comment from one of the reviewers in the first round and it shows how each mission of the GNSS-RO contributes to the GW Ep.

- 4. L148: Here, the vertical resolutions may be replaced by vertical step. Please use "resolution" properly throughout the manuscript for the purpose of more readable and accurate.

  Fixed
- 5. L204-210: "excluding any with vertical wavelengths greater than 12 km", Does this apply on both SABER and GNSS-RO data? How about the shortest vertical wavelengths of GWs derived from SABER and GNSS-RO data?

Yes, it applies to both GNSS-RO and SABER. We set a limit for the highest vertical wavelengths to examine only the short vertical gravity waves due to their importance in interaction with the QBO. The lower limit is defined by the Nyquist sampling limit of each instrument.

6. L223: "eastward peaks of the zonal wind": How about the time step of zonal wind, daily or monthly? It seems that the zonal wind is on a monthly scale as seen from Figure 3. Please clarify this in the text.

Fixed.

7. L281-284: "The horizontal resolution" might be "the horizontal sampling step"? This leads the resolved horizontal wavelength of more than 270 km? Moreover, this is conflict to the statements in L260-264: "however, accurate measurements of the  $\lambda h$  are difficult or impossible for GNSS-RO because the profiles are not regularly spaced". In fact, even at the early missions of GNSS-RO, the RO profiles were not regularly spaced. Please clarify these statements.

The horizontal resolution and the horizontal sampling step are two different concepts, one is a property of the instrument and the other is a choice for the analysis to sample the data. See our response to major comment 3 for further detail.

Based on radio propagation theory as derived by Kursinski et al., 1997, the horizontal line-of-sight resolution is around 270 km. Hence, gravity waves with  $\lambda_h \sim 270$  km in the line of sight are less likely to be detected by GNSS-RO in the along-boresight direction but can be detected in the across-boresight direction and hence the bound is an approximation rather than a hard limit. The upper limit is not well-defined, and fundamentally limited by the minimal detectable phase change in the vertical imposed by the smoothly varying wave along an unknown angle.

Yes, the early missions were not regularly spaced but close enough to be able to estimate the horizontal wavelength of GW using Alexander 1998 method and this has been published by Alexander and Wang 2010 and Faber et al. 2013.

8. L281-288: From these sentences, I get that the upper limit of the vertical wavelength is 12 km for both SABER and GNSS-RO. I guess that the lower limit of the vertical wavelengths is, receptively, 2 km for GNSS-RO and 4 km for SABER. However, the lower limit of the vertical wavelengths shown in Figure A2 is receptively, 4 km for GNSS-RO and 7 km for SABER. I confuse that what is the lower limit of the vertical wavelength, which is used to derive Ep from GNSS-RO and SABER. This is important since the Ep is dependent on the range of the vertical wavelength even for the profiles measured by

**same instrument. Please clarify the lower limit of the vertical wavelength, which is used to derive Ep.**

We did not set a lower limit for the vertical wavelength calculation, but the proportion of very short vertical wavelengths captured by each instrument to the rest of the spectra is very small. The most dominant wavelengths captured by each instrument are the ones shown in figure A2.

**9. L292: Please provide the method of get periodogram.** Fixed.**

10. L339-346: This paragraph provides a statistical test for specific period. However, it is better to the confidence level of the statistical test. For example, the QBO is 2.4 years with 0 FAP. One can infer that the period of 2.5 years has a FAP of below 0.01, which is still a secure result. Due to the variability of QBO period from year to year, a better representation of QBO period might be a statistical distribution with center of ? months and standard deviation of ? months. The significance of periodogram peaks is typically evaluated using the false alarm probability (FAP), which is commonly calculated under the assumption of uncorrelated noise through numerical methods such as bootstrapping or Monte Carlo simulations. FAP indicates the likelihood that a detected peak arises from noise rather than a genuine signal. In our case, the very low FAP values strongly suggest that the observed peaks are indeed due to a true signal. We selected this approach because it is widely used for assessing the statistical significance of periodogram peaks, relies on robust numerical techniques, and provides results that are straightforward for readers to interpret.

**11. Section 5.1: The range of vertical wavelengths of GW, which is used to derive Ep should also be clarified.**

Fixed (L426).

---

## Author Response (AR3)

**Response to Reviewers**

We thank the reviewers for their comments and the time spent preparing them.

**Reviewer 1**

**Major comments:**

Thanks for your responses. However, I would like to see the positive responses to my comments in the last two version. Moreover, please provide directect proof(s) to support your idea and purpose in the abstract. If the proof are at other places of the text, please provide them clearly in the responses or the line numbers in the text.

Beside the unresolved comments in the last two versions, a new comments is that the title may be confined in the stratosphere since the temperature observed by GNSS-RO cannot cover the mesosphere and lower thermosphere where SABER covered.

We have completely rewritten the abstract and title of the manuscript, as requested by Reviewer 1 and subsequently clarified in writing with the Editor. Additionally, we have made the main points of the study clearer in the latest version as follow:

- 1. The title is now confined to the stratosphere only.
- 2. We have highlighted the importance of the study and its critical timing since SABER will no longer be running after 2025 and there is no planned replacement (L7-10).
- 3. We have also referred to the main differences between SABER and GNSS-RO which make their results vary and described how to tackle these differences in order to get as close agreement as possible (L11-12).
- 4. We have proposed using a previously demonstrated method to compare the two measurements as closely as possible by using synthetic data from a high-resolution global model. This method has been tested in a published study and showed a very high accuracy in sampling the satellite data using model data (Wright & Hindley 2018). The details of this method and the results are shown in section 5.
- 5. The main result of the study is clarified in L17 -19.
- 6. The main question of the study is illustrated and answered clearly by adding Figure 12 after the reviewer suggestion in round 1.

---

## Author Response (AR4)

**Response to Reviewers**

We thank the reviewers for their comments and the time spent preparing them.

**Reviewer 1**

**Minor comments:**

1. L196-199: One cannot get any information on the method of removing large-scale variations from the scentences in these lines. For the completeness of the manuscript, please sumarize the key idea/physics behind your new method here (Hindley et al., 2022).

Thank you. We have added some paragraphs to clarify the method (L197 - L212) and edited Fig. 1 which shows a flowchart of the steps for the method.

2. L214-223: The large-scale backgrounds are removed by a daily weighted sine fitting routine. The waves with zonal wavenumbers from 0-9 represent the background. Since the vertical wavelength of gravity and Kelvin waves could be similar (L203), how to ensure that the Kelvin waves are removed by the method proposed here (i.e., daily weighted sine fitting routine)?

Dominant Kelvin waves of the lower stratosphere are generally of zonal wavenumbers 1 and 2 and to have periods in the range of 10-20 days. Faster Kelvin waves with period 6 -10 days, and ultrafast Kelvin waves with periods of 2-5 days in the stratosphere. Our method is using a fitting sinusoidal for days with a window of 1 day, hence, excluding Kelvin waves by period. The method is also removing Kelvin waves on the zonal scale since they exclude any waves with wave numbers from 0 to 9.

Yes, Kelvin waves have similar vertical wavelength as gravity waves and there might be a very small contribution from them in the signal, but the zonal removal is the only way to exclude them and the major signature in the output is due to gravity waves.

---

## Author Response (AR5)

**A point-by-point response to the Editor:**

For the next revision please use the initials instead of the full names of the authors for the section "Author's contribution" (e.g. M. Almowafy -> MA, etc.).

Corrected (L720).